**Marine anoxia initiates giant sulfur-bacteria mat proliferation and associated changes in**
**benthic nitrogen, sulfur, and iron cycling in the Santa Barbara Basin, California**
**Borderland**
David J. Yousavich[1*], De'Marcus Robinson[2], Xuefeng Peng[3], Sebastian J. E. Krause[1,4], Frank
Wenzhöfer[5,6,7], Felix Janssen[5,6], Na Liu[8], Jonathan Tarn[8], Frank Kinnaman[8], David L. Valentine[8],
Tina Treude[1,2*]
[1]Department of Earth, Planetary, and Space Sciences, University of California Los Angeles, 595 Charles E.
Young Drive East, Los Angeles, CA 90095, USA
[2]Department of Atmospheric and Oceanic Sciences, University of California Los Angeles, Math Science
Building, 520 Portola Plaza, Los Angeles, CA 90095, USA
[3]School of Earth, Ocean, and Environment, University of South Carolina, 701 Sumter Street, EWS 617,
Columbia, SC 29208, USA
[4]Earth Research Institute, 6832 Ellison Hall, University of California Santa Barbara, Ca 93106-3060
[5]HGF-MPG Joint Research Group for Deep-Sea Ecology and Technology, Alfred-Wegener-Institute,
Helmholtz-Center for Polar and Marine Research, Am Handelshafen 12, 27570 Bremerhaven, Germany
[6] HGF-MPG Joint Research Group for Deep-Sea Ecology and Technology, Max Planck Institute for Marine
Microbiology, Celsiusstrasse 1, 28359 Bremen, Germany
[7]Department of Biology, DIAS, Nordcee and HADAL Centres, University of Southern Denmark, 5230 Odense
M, Denmark
[8]Department of Earth Science and Marine Science Institute, University of California, Santa Barbara, CA
93106, USA
**Correspondence:** David Yousavich (yousavdj@ucla.edu), Tina Treude (ttreude@g.ucla.edu)

**Abstract**

The Santa Barbara Basin naturally experiences transient deoxygenation due to its unique geological setting in the Southern California Borderland and seasonal changes in ocean currents. Long-term measurements of the basin showed that anoxic events and subsequent nitrate exhaustion in the bottom waters have been occurring more frequently and lasting longer over the past decade. One characteristic of the Santa Barbara Basin is the seasonal development of extensive mats of benthic nitrate-reducing sulfur-oxidizing bacteria, which are found at the sediment-water interface when the basin's bottom waters reach anoxia but still provide some nitrate. To assess the mat's impact on the benthic and pelagic redox environment, we collected biogeochemical sediment and benthic flux data in November 2019, after anoxia developed in the deepest waters of the basin and dissolved nitrate was depleted (down to 9.9 µM). We found that the development of mats was associated with a shift from denitrification to dissimilatory nitrate reduction to ammonium. The zone of sulfate reduction appeared near the sediment-water interface in sediment hosting these ephemeral white mats. We found that an exhaustion of iron oxides in the surface sediment was an additional prerequisite for mat proliferation. Our research further suggests that cycles of deoxygenation and reoxygenation of the benthic environment result in extremely high benthic fluxes of dissolved iron from the basin's sediment. This work expands our understanding of nitrate-reducing sulfur-oxidizing mats and their role in sustaining and potentially expanding marine anoxia.

**Introduction**

Naturally occurring low-oxygen waters in the ocean are commonly observed below the ocean's mixed layer where respiration consumes oxygen faster than it is produced or ventilated. When low oxygen conditions occur along the western continental shelf in regions susceptible to upwelling events and/or undergoing eutrophication, organic matter remineralization can frequently drive oxygen concentrations to hypoxic ($O_2 < 63$ μM) (Middelburg and Levin, 2009) and/or anoxic levels ($O_2 < 3$ μM) (Fossing et al., 1995; Canfield et al., 2010). These areas are usually referred to as Oxygen Minimum Zones (OMZs). In the water column of OMZs, nitrogen reduction becomes an important mechanism for organic matter remineralization (Ward et al., 2009). OMZs within coastal basins that experience seasonal changes in upwelling can experience anoxic and nitrate reducing conditions that extend to the benthic environment, especially when high productivity and associated organic matter export coincide with seasonal patterns of physical mixing. This fundamental change in the redox conditions at the sediment-water interface encourages elevated rates of anaerobic microbial processes and can promote organic matter preservation in the sediments (Middelburg and Levin, 2009; Treude, 2011), though a recent study suggests a thin reactive surface layer can provide high rates of organic matter degradation in anoxic environments (Van De Velde et al., 2023). Persistent anoxia in these coastal OMZ can lead to huge releases of sulfide (up to 13.7 mmol $m^{-2}$ $d^{-1}$) and ammonium (up to 21.2 mmol $m^{-2}$ $d^{-1}$) into the water column (Sommer et al., 2016).

The Santa Barbara Basin (SBB) is an example of one of these coastal OMZs that experiences seasonal deoxygenation. Drastic changes in water column oxygenation and seafloor redox

conditions drive complex changes in benthic biogeochemistry and microbiology, evidenced most
clearly by the development of thick, expansive mats of giant sulfur-oxidizing bacteria (GSOB)
on the SBB seafloor (Bernhard et al., 2003; Prokopenko et al., 2006; Valentine et al., 2016;
Kuwabara et al., 1999). A 2016 survey of the basin identified a vast GSOB mat spread over 1.6
contiguous km, confined between 487 and 523 km in the SBB depocenter where conditions were
anoxic but not depleted of $NO_3^-$ (Valentine et al., 2016). These GSOB mats have been noted
previously in the SBB benthos, appearing at times of anoxia and disappearing when oxygen is
present in the bottom water (Reimers et al., 1996; Kuwabara et al., 1999). Similar GSOB mats
have been identified in other transiently deoxygenated OMZs such as the Peruvian/Chilean coast
(Sommer et al., 2016; Schulz et al., 1996; Zopfi et al., 2001; Høgslund et al., 2009). The
chemoautotrophic bacteria that constitute the bulk of GSOB mats (typically *Thioploca* and/or
*Beggiatoa*) utilize sulfide as an electron donor and $O_2$ or $NO_3^-$ as a terminal electron acceptor
(Jørgensen and Nelson, 2004). Some GSOB can hyperaccumulate $NO_3^-$ in cell vacuoles up to
500 mM (Fossing et al., 1995) and use this $NO_3^-$ reserve to oxidize sulfide that diffuses from the
underlying sediment to perform their metabolism. (Huettel et al., 1996; Mußmann et al., 2003;
Sayama, 2001).

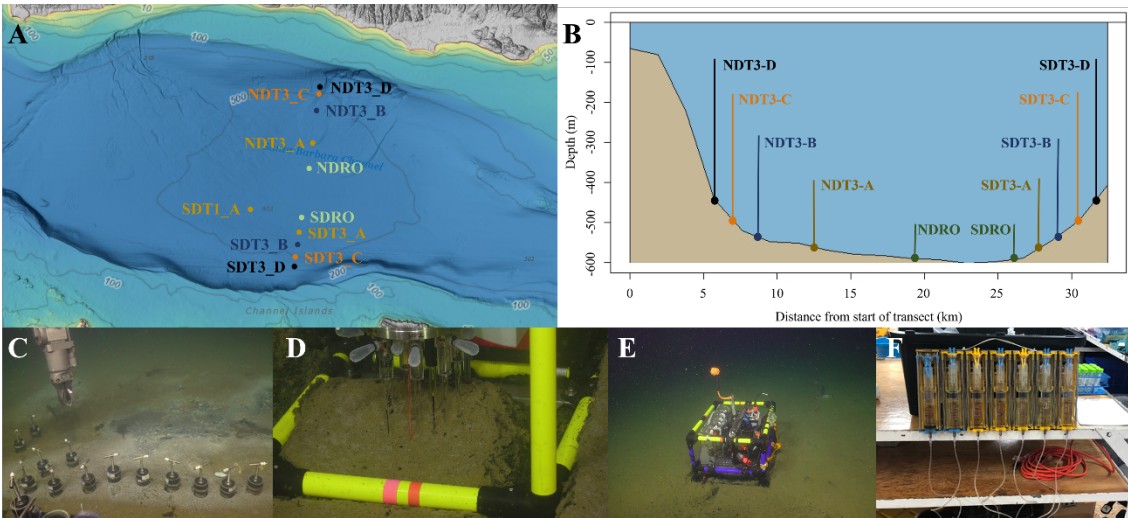

**Figure 1. Maps of sampling locations in the Santa Barbara Basin and photographs of deployed equipment:** (A) bathymetric map of the Santa Barbara Basin with locations of all sampled stations; (B) cross-section of the Santa Barbara Basin with locations of all sampled station; (C) sediment push coring with ROV arm; (D) sediment microprofiler; (E) benthic flux chamber; (F) closeup of a syringe system from a benthic flux chamber. The map in (A) was generated using the Bathymetric Data Viewer provided by the National Centers for Environmental Information.

The activity of GSOB mats contribute significantly to element cycling in benthic marine environments with large effects on biogeochemical conditions in the bottom water. Isotopic measurements of $^{15}N/^{14}N$ and $^{18}O/^{16}O$ from $NO_3^-$ in the SBB water column suggest that benthic organisms are responsible for approximately 75% of the total $NO_3^-$ reduction in the SBB (Sigman et al., 2003). Other studies found that GSOB mats inhibit the diffusion of $NO_3^-$ into sediments via hyper-accumulation in vacuoles thereby creating conditions ideal for bacterial heterotrophic sulfate reduction beneath them (Fossing et al., 1995; Zopfi et al., 2001). These studies suggest that GSOB mats in the SBB may be responsible for the majority of $NO_3^-$ consumption in the basin rather than water-column microbes. Additionally, GSOB mats have been reported to deplete $NO_3^-$ via dissimilatory nitrate reduction to ammonia (DNRA) in the anoxic bottom water of the Peruvian OMZ (Dale et al., 2016) and in the hypoxic transition zone

in the Eastern Gotland Basin of the Baltic Sea (Noffke et al., 2016). By contrast, benthic
microbial communities in the hypoxic (42 μM) Mauritanian OMZ perform canonical
denitrification instead (Dale et al., 2014). The contrast between the Peruvian and Mauritanian
OMZ suggests that bottom- water anoxia triggers the appearance of GSOB mats, and that DNRA
is more prevalent where GSOB mats are present.

The rapid accumulation and consumption of $NO_3^-$ by GSOB mats has ramifications for the redox
conditions in the sediment underneath. The depletion of $NO_3^-$ and shallowing of the nitracline
could promote high rates of sulfate reduction in the sediment underneath the GSOB mat. In
return, the sulfate reduction zone exists close to the sediment-water interface, providing the
GSOB mat with readily accessible sulfide. If a metabolic feedback loop is then established
between sulfur-oxidizing bacteria at the sediment-water interface and sulfate-reducing bacteria in
the sediment, increased $NO_3^-$ loss from the water column and spreading of sulfidic conditions in
SBB sediment is expected. With these mats being potentially crucial to nitrogen and sulfur
cycling in sediments underlying OMZs, their biogeochemical transformations and ergo effect
upon basin redox conditions are critically important to understanding element cycling in the
SBB. Such gained knowledge would have additional benefits for predicting biogeochemical
feedbacks to the projected expansion of oceanic oxygen deficiency, in the SBB and in OMZs
more general, as a result of global change (Stramma et al., 2008).

Utilizing in-situ technologies, sediment porewater extraction, solid phase analyses, and
radiotracer techniques, this study aims to answer the following overarching questions: (1) Which
environmental conditions initiate and sustain the proliferation of GSOB mats? (2) Which
biogeochemical transformations occur in the sediment underneath these mats? (3) What role do
the mats play in the increasingly prevalent anoxic and nitrate-depleted condition found in the
SBB? These investigations represent the first basin-wide geochemical characterization of the
Santa Barbara Basin which hosts the largest as-of-yet mapped GSOB mat in the world's oceans.
It is the first suite of in-situ flux measurements carried out in the SBB, which is unique to other
heavily studied marine settings (e.g., Eastern Gotland Basin, Peruvian upwelling zone) in that it
is an oceanic basin within an upwelling zone. The results presented here also provide
geochemical context for a number of other related investigations in the SBB (Robinson et al.,
2022; Xuefeng Peng et al., 2023; Peng et al., 2023) as well as the first measurements in a multi-
year study of biogeochemical changes in response to warming waters and increased stratification
on the California coast.

## 2. Materials and Methods

### 2.1 Study Site

The Santa Barbara Basin (SBB) is a coastal basin in the California Borderland with an approximate maximum depth of 600 m characterized by a seasonally anoxic water column (Sverdrup and Allen, 1939; Sholkovitz and Gieskes, 1971). The transform boundary along the California Borderland heavily affects the geomorphology of basins in this region; these basins become twisted as the plates rub against each other and form a series of "bathtubs" blocked by sills and seamounts off the coast of California. The SBB is bordered by the California coast in the north, the Channel Islands in the south, the Santa Monica basin to the east, and the Arguello Canyon to the west. A sill to the west of the basin at around 475 m depth (Fig. 1) prohibits most water transfer between the Santa Lucia Slope and the deeper waters of the SBB (Sholkovitz and Gieskes, 1971). The highly productive surface waters in the basin provide ample organic matter to the basin's water column, encouraging strong remineralization processes below the euphotic zone, which can induce anoxia below the sill depth, with typically less than 1 $\mu$mol $O_2$ $L^{-1}$ (Sholkovitz, 1973; Emery et al., 1962; Thunell, 1998; Emmer and Thunell, 2000). Benthic faunal distribution within the basin is tightly correlated with this sill depth and related oxygen conditions; below the sill, the sea snail *Alia permodesta* is the most commonly found benthic fauna, while sea stars, sea urchins, and other echinoderms increase in density above the sill (Myhre et al., 2018). During upwelling events (usually in Spring), oxygenated waters from the California Current spill over the western sill and ventilate the SBB, reportedly increase bottom water oxygen concentrations to approximately 20 $\mu$mol $O_2$ $L^{-1}$ (Goericke et al., 2015). SBB water-column oxygen and nitrogen concentrations have been evaluated through a longitudinal survey by the California Cooperative Oceanic Fisheries Investigations (Calcofi) with data

starting in the 1950's . The data collected by this survey shows increasing durations of anoxia
and fixed nitrogen loss in the basin with the SBB becoming completely nitrate-depleted below
the sill at least three times between 2012 and 2017 (https://calcof.iorg/data/).

**2.2 Benthic sediment sampling and instrument deployment**
Sediment samples were taken between 30 October and 11 November 2019 during an expedition
aboard the research vessel *(R/V) Atlantis* equipped with the remote operated vehicle (ROV)
Jason. Samples were taken at stations along a bimodal, north-south transect through the
depocenter of the SBB, as well as one station on a separate transect. Details of sampling stations
can be seen in Fig. 1A and 1B. Briefly, depocenter stations are labeled as NDRO and SDRO
(northern and southern depocenter radial origin, respectively). The remaining stations are named
for the cardinal direction (north vs. south) and the transect number (e.g., SDT1-A is on transect 1
while SDT3-A is on transect 3). As station depth decreases, the alpha suffix increases (e.g.,
NDT3-A is deeper than NDT3-B, etc.).

ROV Jason conducted sediment push coring and deployed automated benthic flux chambers
(BFC) and microprofilers at each station. Bottom water oxygen concentration was determined
using an Aanderaa 4831 oxygen optode (Aanderaa Instruments, Bergen, Norway) installed on
the ROV. Optical modems (Luma 250LP, Hydromea, Renens, Switzerland) installed on the ROV
and the BFC and microprofilers were used to transmit deployment settings and start/terminate
measurements of the instruments. Multiple push cores (polycarbonate, 30.5 cm length, 6.35 cm
inner diameter) per sampling station were retrieved during ROV Jason deployments (Fig. 1C).
Replicate cores from each station were transferred to an onboard 6°C cold room upon recovery
aboard the ship and subsampled for either solid phase analyses, porewater geochemistry, or
radiotracer experiments.

**2.3 Sediment Core Sub-Sampling**
Two replicate ROV push cores that were collected near each other at each station were processed
under a constant argon flow to protect redox-sensitive species. Cores were sectioned in 1-cm
increments up to 10 cm followed by 2-cm increments. Note, sediments from the NDT3-B station
were sliced in 2-cm increments. Sediment subsections were transferred into argon-filled 50-mL
conical centrifuge tubes. Sediment samples were centrifuged at 2300 x g for 20 minutes. The
centrifugate was subsampled unfiltered as fast as possible (to avoid contaminations with oxygen)
for porewater analyses. Solid phase cores were sectioned similar to porewater cores and sub-
sampled for sediment density, porosity, and organic matter content. A 10 mL cut-off plastic
syringe was used to collect 6 mL of sediment into pre-weighed plastic vials (15 mL snap-cap
vials) and stored in the dark at 4°C for sediment porosity and density analysis. Two-mL
microcentrifuge tubes were filled with sediment from each depth interval and stored at -30°C for
sediment organic matter analyses. One ROV push core per station was sub-sampled with a
miniaturized push core (length 20 cm, inner diameter 2.6 cm) and taken immediately to the
shipboard radioisotope van for radiotracer experiments (see section 2.5).

**2.4 Sediment Porewater Geochemistry**
Concentrations of porewater sulfide (Cline, 1969), $NH_4^+$, $PO_4^{3-}$, and $Fe^{2+}$ (Grasshoff et al., 1999)
were determined shipboard with a Shimadzu UV-Spectrophotometer (UV-1800). Detection
limits for sulfide, $NH_4^+$, $PO_4^{3-}$, and $Fe^{2+}$ were 1 µM. Subsamples (2 mL) for porewater $NO_3^-$ and
$NO_2^-$ concentrations were stored in 2-mL plastic vials with an O-ring, frozen shipboard at -30°C
and analyzed back at the home laboratory on the same spectrophotometer using the method
following (García-Robledo et al., 2014). The detection limit for $NO_3^-$ and $NO_2^-$ was 0.5 µM.
Samples for porewater DIC were preserved shipboard with 5 µL saturated $HgCl$ in headspace
free glass vials and stored at 4°C for later analysis following (Hall and Aller, 1992). DIC
detection limit was 0.1 mM. Total alkalinity was determined shipboard using direct titration of
500 µL of pore water with 0.01M Titrisol® HCl (Pavlova et al., 2008). The analysis was
calibrated using IAPSO seawater standard, with a precision and detection limit of 0.05 meq $L^{-1}$.
Subsamples (1 mL) for sulfate and chlorinity were stored in 2-mL plastic vials with an O-ring,
frozen shipboard at -30°C and later measured in the lab using a Metrohm 761 ion chromatograph
with a methodological detection limit of 30 µM (Dale et al., 2015).

**2.5 Solid Phase Analyses**
Porosity/Density samples were collected in pre-weighed plastic vials and dried at 50°C for up to
96 hr until the dry weight was stable. Sediment porosity was calculated by taking the difference
between wet and dry sediment weight and divided by the volume of the wet sediment. Sediment
density was calculated by dividing the wet sediment weight by its volume. Treatment of
sediment subsamples for total organic carbon (TOC), total organic nitrogen (TON), and organic
carbon isotope composition ($\delta^{13}C$) were modified from (Harris et al., 2001) and sent to the
University of California Davis Stable Isotope Facility for analysis using Elemental Analyzer –
Isotope Ratio Mass Spectrometry. TOC and TON were calculated based on the sample peak area
corrected against a reference material (alfalfa flour). Limit of quantification based on peak area
was 100 µg C with an uncertainty of ± 0.2 ‰ for $\delta^{13}C$.

## 2.6 Sulfate Reduction

To determine ex-situ microbial sulfate reduction rates, whole round sub-cores were injected with
10 µL carrier-free $^{35}$S-Sulfate radiotracer (dissolved in water, 200 kBq, specific activity 37 TBq
mmol$^{-1}$) into pre-drilled, silicon-filled holes at 1-cm increments according to (Jørgensen, 1978).
These sub-cores were incubated at 6°C in the dark for 6-8 hours. Incubations were stopped by
slicing sediment cores in 1-cm increments into 50-mL centrifuge tubes filled with 20-mL zinc
acetate (20% w/w) and frozen at -20°C until analysis at the land-based laboratory. Microbial
activity in controls was terminated with zinc acetate (20 mL of 20% w/w) before the addition of
radiotracer and subsequent freezing. Lab-based analysis of sulfate reduction rates were
determined following the cold-chromium distillation procedure (Kallmeyer et al., 2004).

## 2.7 Benthic In-Situ Investigations

Per station, one to three microprofiler (Fig. 1D) and three BFC (Fig. 1E) deployments were
carried out by the ROV Jason at the seafloor. Construction, deployment and operation of
automated microprofilers and BFCs followed those described in (Treude et al., 2009). The
microprofiler deployed in this study represents a modified, miniaturized version of the
instrument described in (Gundersen and Jørgensen, 1990) that was constructed specifically for
use by ROV. Microprofilers were outfitted with three $O_2$-microelectrodes (Glud et al., 2000),
two pH-microelectrodes (Revsbech and Jørgensen, 1986), two $H_2S$-microelectrodes
(Jeroschewsky et al., 1996), and one conductivity sensor to determine the position of the
sediment-water interface relative to the tips of the microelectrodes. Concentrations of oxygen
and sulfide, as well as pH were each calculated from microelectrode readings and averaged for
the respective sites where replicates existed.

The BFC consisted of a frame equipped with a cylindrical polycarbonate chamber (inner
diameter = 19 cm) with its lower portion sticking out of the frame. The upper side of the
chamber was closed by a lid containing a stirrer (Type K/MT 11, K.U.M., Kiel, Germany),
oxygen optodes (Type 4330, Aanderaa Data Instruments, Bergen Norway and Hydroflash,
Contros/Kongsberg Maritime, Kongsberg, Norway), a conductivity sensor (type 5860, Aanderaa
Data Instruments), and a valve. Prior to insertion into the sediments, the chambers were held
upside down by the ROV manipulating arms within approximately 10 m of the seafloor and
moved back and forth to make sure that water from shallower depth that may have been trapped
was replaced by bottom water. Chamber incubations lasted between 240 and 390 minutes. Each
BFC was outfitted with a custom-built syringe sampler containing seven syringes that were
connected by tubes to sampling ports in the upper wall of the chambers (Fig. 1F):  one injection
syringe and six sampling syringes that were fired at regular time intervals over the time course of
the deployment. The injection syringe contained de-ionized water and the reduction in salinity in
the overlaying water after salinity readings stabilized (i.e., full mixing was achieved) 10-30 min
after injection was used to determine BFC volumes (Kononets et al., 2021). Samples obtained
from the overlaying water of the BFC were examined for the same geochemical constituents as
described above (section 2.4). Benthic fluxes of $NO_3^-$, $NH_4^+$, $PO_4^{3-}$, and $Fe^{2+}$ were calculated as
follows:

$J = \frac{\Delta c}{\Delta t} * \frac{V}{A}$                                                                (EQ # 2)

Where J is the flux in mmol m$^{-2}$ d$^{-1}$, $\Delta$C is the concentration change in mmol m$^{-3}$, $\Delta$t is the time
interval in d, V is the overlying water volume in m$^3$, and A is the surface area of the sediment
covered by the benthic flux chamber in m$^2$. An average flux within BFC's was calculated for
stations of similar depth. One chamber per site contained $^{15}$N-NO$_3^-$ in the injection syringe for in-
situ nitrogen cycling experiments. Results are reported from two of these chambers (SDRO and
NDT3-D) and all $^{15}$N-NO$_3^-$ chambers were excluded from benthic flux calculations (see next
section).

**2.8 In Situ $^{15}$N Incubations**
Two hundred μmol of $^{15}$N-labeled potassium nitrate (99% $^{15}$N; Cambridge Isotopes) was injected
into the $^{15}$N incubation chamber at each site to obtain a final concentration of ~50 – 100 μM $^{15}$N-
labeled nitrate. Nitrate was amended at this level to prevent its depletion before the last sampling
time point (Valentine et al., 2016). Samples for δ$^{15}$N analysis were preserved by filling a pre-
vacuumed 12-ml exetainer vial with 0.1 ml 7M zinc chloride as a preservative. Another aliquot
(~12 ml) of seawater for ammonium isotope analysis (see section 2.7.2) was filtered through 0.2
μm syringe filters and stored frozen. Prior to analyzing the samples in 12-ml exetainer vials, 5
mL of sample was replaced with ultra-high purity helium to create a headspace. The
concentration and δ$^{15}$N of dissolved N$_2$ and N$_2$O was determined using a Sercon CryoPrep gas
concentration system interfaced to a Sercon 20-20 isotope-ratio mass spectrometer (IRMS) at the
University of California Davis Stable Isotope Facility.

**2.9 Ammonium Isotope Analyses**
The production of $^{15}NH_4^+$ in seawater samples was measured using a method adapted from
(Zhang et al., 2007) and described previously by (Peng et al., 2016). In brief, $NH_4^+$ was first
oxidized to $NO_2^-$ using hypobromite ($BrO^-$) and then reduced to $N_2O$ using an acetic acid-azide
working solution (Zhang et al., 2007). The $\delta^{15}N$ of the produced $N_2O$ was determined using an
Elementar Americas PrecisION continuous flow, multicollector, isotope-ratio mass spectrometer
coupled to an automated gas extraction system as described in (Charoenpong et al., 2014).
Calibration and correction were performed as described in (Bourbonnais et al., 2017). The
measurement precision was ±0.2 ‰ for $\delta^{15}N$. Depending on the in-situ ammonium
concentration, the detection limit for total $NH_4^+$ production rates ranged between 0.006 and
0.0685 mmol m$^{-2}$ d$^{-1}$.

## 3. Results

### 3.1 Bottom water conditions

$O_2$ and $NO_3^-$ concentrations in the bottom water along the transects can be seen in Table 1. $O_2$ concentrations below detection as determined by the ROV sensor could in some cases be considered to represent anoxia ($0\ \mu M\ O_2$) based on a set of different analytical methods (see discussion section 4.1). Bottom water solute concentrations (as defined by the average $T_0$ concentration in BFC at each site) can be seen in Suppl. Figs. 1-4. Bottom water $NO_3^-$ concentrations roughly decreased with station depth (e.g., 28 $\mu M$ at NDT3-D vs. 19 $\mu M$ at NDRO). Bottom water $NO_2^-$ concentrations were below detection at all stations. Bottom water $NH_4^+$ concentrations were 9 $\mu M$ at NDRO and 13 $\mu M$ at SDRO and below detection in shallower stations. Bottom water $PO_4^{3-}$ concentrations roughly increased with increasing basin depth (e.g., 2 $\mu M$ at SDT3-D vs. 7 $\mu M$ at SDRO). Finally, $Fe^{2+}$ was 2 and 5 $\mu M$ at the NDRO and SDRO stations, respectively and below detection at all shallower stations.

### 3.2 Sediment characteristics

Photographs of sediment cores with a depth scale are shown below Table 1. Sediment colors were classified according to (Hossain et al., 2014). Cores from the shallowest (D) stations were uniformly reddish in color with small pockets of black. The sediment color changed with station depth, transitioning from a reddish color in the shallowest stations to predominantly black with reddish laminations at the depocenter stations. The band of black sediment appeared at approx. 8 cm sediment depth in the C-station cores and became progressively more ubiquitous with station depth. Notably, NDT3-C sediment (Table 1B) contained black bands from approx. 6-14 cm sediment depth, while SDT3-C sediment (Table 1J) had a much narrower band around 8-10 cm.

Sediment cores from shallower stations (D and C stations) contained signs of bioturbation (e.g.,
u-shaped burrows) and, in some cases, contained visible macrofauna, such as polychaetas and
mollusks. Deeper in the basin (A and depocenter stations) no signs of bioturbation were detected,

**Table 1.** Station details and photos of representative ROV push cores taken at each station. Mat presence (Y =
yes, N = no) was determined visually. Station water depth and oxygen concentration were determined by sensors
attached to ROV Jason (bdl = below detection limit (<3 µM O$_2$)). Anoxia was confirmed by additional methods
(see discussion section 4.1). Latitude and longitude were determined by triangulation between the ROV and the
ship. Bottom water nitrate concentration was derived from an average of benthic flux chamber nitrate
measurements at time 0 for each station (chambers with no calculatable flux and [15]N-nitrate addition excluded).
Note, benthic flux chambers were not deployed at SDT1-A. Photographs show the sediment-water interface
(SWI; top part) and each sediment core in full length (lower part).

| Parameter | NDT3-D | NDT3-C | NDT3-B | NDT3-A | NDRO | SDRO | SDT1-A | SDT3-A | SDT3-B | SDT3-C | SDT3-D |
|---|---|---|---|---|---|---|---|---|---|---|---|
| Mat Present | N | N | N | Y | Y | Y | Y | Y | N | N | N |
| Depth (m) | 447 | 498 | 537 | 572 | 580 | 586 | 573 | 571 | 536 | 494 | 447 |
| Latitude (°) | 34.363 | 34.353 | 34.333 | 34.292 | 34.262 | 34.201 | 34.212 | 34.184 | 34.168 | 34.152 | 34.142 |
| Longitude (°) | -120.015 | -120.016 | -120.019 | -120.026 | -120.031 | -120.044 | -120.116 | -120.047 | -120.053 | -120.050 | -120.052 |
| Oxygen (µM) | 8.7 | 5.2 | 12.2 | 9.2 | 0.0 | 0.0 | 0.0 | 0.0 | 1.8 | 3.1 | 9.6 |
| Nitrate (µM) | 27.3 | 26.0 | 11.5 | 24.4 | 18.5 | 9.9 | | 20.4 | 20.6 | 16.3 | 28.0 |

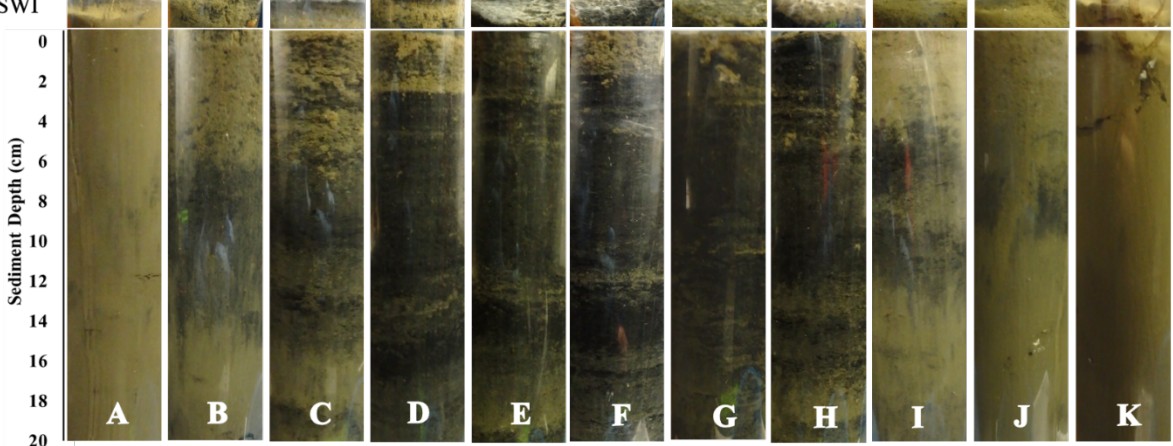


and the sediment-water interface was colonized by patches of white GSOB mats. Spherical cells
(given the moniker 'ghost balls') were found mixed amongst giant sulfur bacteria filaments
within the top 0-1 cm of sediment at NDRO (Suppl. Fig. 7). These unknown species had similar
morphological characteristics to the species *Thiomargarita namibiensis* (Schulz et al., 1999;
Schulz and Schulz, 2005) containing a translucent cell with sulfur granules giving them a ghostly
white appearance. A small sample of cells (n = 8) were measured, featuring diameters between
48.0 and 99.6 μm, amounting to an average biovolume of $2.5 \times 10^5$ μm$^3$, compared to *T.*
*namibiensis* with a cell diameter usually between 100-300 μm (Schulz et al., 1999).
**Table 2.** Sediment solid phase data: porosity, density, total organic carbon (TOC), total organic nitrogen (TON),
C:N ratio, and δ13C. All data were averaged for the top 0-19 cm sediment, except NDT3-C (17 cm), NDT3-A
(11 cm) and SDRO (7 cm), where the core length was shorter. Integrated sulfate reduction rates (iSRR) were
integrated over 0-14 cm sediment depth. No sulfate reduction rates are available for NDT3-B, SDT3-A, and
SDT3-B; rates were not integrated for SDRO due to missing surface samples.

| Parameter | NDT3-D | NDT3-C | NDT3-B | NDT3-A | NDRO | SDRO | SDT1-A | SDT3-A | SDT3-B | SDT3-C | SDT3-D |
|---|---|---|---|---|---|---|---|---|---|---|---|
| Porosity | $0.79 \pm 0.03$ | $0.81 \pm 0.04$ | $0.86 \pm 0.04$ | $0.88 \pm 0.03$ | $0.88 \pm 0.04$ | $0.87 \pm 0.03$ | $0.88 \pm 0.03$ | $0.86 \pm 0.04$ | $0.85 \pm 0.04$ | $0.82 \pm 0.04$ | $0.78 \pm 0.04$ |
| Density | $1.21 \pm 0.07$ | $1.16 \pm 0.08$ | $1.06 \pm 0.08$ | $1.05 \pm 0.04$ | $1.06 \pm 0.03$ | $1.04 \pm 0.03$ | $1.11 \pm 0.23$ | $1.05 \pm 0.05$ | $1.12 \pm 0.06$ | $1.22 \pm 0.05$ | $1.22 \pm 0.03$ |
| TOC (%) | $2.9 \pm 0.5$ | $2.5 \pm 0.5$ | $3.6 \pm 0.5$ | $3.1 \pm 0.4$ | $3.3 \pm 0.4$ | $3.5 \pm 0.4$ | $4.5 \pm 0.5$ | $3.2 \pm 0.0$ | $3.6 \pm 0.6$ | $3.6 \pm 0.8$ | $3.3 \pm 0.5$ |
| TON (%) | $0.3 \pm 0.1$ | $0.5 \pm 0.1$ | $0.4 \pm 0.1$ | $0.4 \pm 0.1$ | $0.4 \pm 0.0$ | $0.4 \pm 0.1$ | $1.0 \pm 0.1$ | $0.4 \pm 0.0$ | $0.4 \pm 0.1$ | $0.4 \pm 0.1$ | $0.4 \pm 0.1$ |
| C:N Ratio | $8.9 \pm 0.2$ | $8.7 \pm 0.5$ | $8.5 \pm 0.5$ | $8.2 \pm 0.2$ | $8.2 \pm 0.4$ | $8.0 \pm 0.2$ | $8.6 \pm 0.8$ | $8.3 \pm 0.6$ | $8.3 \pm 0.3$ | $8.7 \pm 0.3$ | $8.5 \pm 0.2$ |
| δ$^{13}$C (‰) | $-22.4 \pm 0.3$ | $-22.4 \pm 0.4$ | $-22.2 \pm 0.4$ | $-22.1 \pm 0.2$ | $-22.1 \pm 0.2$ | $-22.0 \pm 0.3$ | $-21.3 \pm 0.7$ | $-22.1 \pm 0.4$ | $-22.0 \pm 0.2$ | $-21.9 \pm 0.2$ | $-22.0 \pm 0.1$ |
| Integrated SRR (mmol m$^{-2}$ d$^{-2}$) | 2.9 | 3.8 | | 2.7 | 4.1 | | 2.9 | | | 1.7 | 1.9 |

B station cores contained sporadic GSOB filaments slightly deeper in the sediment (approx. 2-4
cm sediment depth). Sediment solid phase parameters (averaged over the entire sediment core
depth) can be seen in Table 2. Average sediment porosity increased with basin depth (e.g., from
0.79 at NDT3-D to 0.88 at NDRO). TOC, TON, the C/N ratio, and the δ$^{13}$C isotopic signature of
organic carbon remained relatively constant (2.5 – 4.5%, 0.1 – 0.4%, 8.0 – 8.7 and 21.3 –
22.4 ‰, respectively) over all stations.

**3.3 Sediment porewater geochemistry**
Total alkalinity (Figs. 2 A-E & 3 A-F) increased steadily with sediment depth at all stations
starting with, on average, 2.4 mM in the core supernatant reaching a maximum at the respective
deepest sediment sample (20 cm). Porewater alkalinity and DIC also increased with basin depth
(Figs. 2 A-E & 3 A-F) indicating that total alkalinity was dominated by the carbonate system.
Porewater DIC was, on average, 2.2 mM in the core supernatant and reached maximum
concentrations at the deepest sediment depth (20 cm) at most stations.

Porewater $PO_4^{3-}$ profiles (Figs. 2 A-E & 3 A-F) were markedly different between the depocenter
and shallower C and D stations. Porewater $PO_4^{3-}$ concentrations in the depocenter and A stations
generally increased with sediment depth but several profiles (NDT3-C, NDT3-A, SDRO, SDT1-
A) remained unchanged or decreased deeper in the sediment (starting at approx. 10 cm). The
profiles in C and D stations showed a peak in $PO_4^{3-}$ concentrations near the sediment-water
interface, particularly in the northern basin. Below 2 cm, $PO_4^{3-}$ decreased with sediment depth,
but sometimes showed a second small peak deeper in the sediment (12-14 cm at NDT3-D and
10-12 cm at SDT3-D).

Porewater $NH_4^+$ concentrations (Figs 2 & 3 A-E) showed trends often similar to alkalinity and
DIC; $NH_4^+$ concentrations increased downcore and were higher at depocenter than at D stations
(e.g., 370 and 91 µM at 20 cm for SDRO and SDT3-D, respectively). Porewater $NO_2^-$ (Suppl.
Table 1) and $NO_3^-$ (Figs. 2 F-J & F G-L) concentrations were at or near zero below 2 cm at every
station, except at SDRO and NDT3-A where large peaks in $NO_3^-$ (376 and 81 µM, respectively)
and $NO_2^-$ (37 and 5 µM, respectively) occurred in the top 1 cm.

Porewater $Fe^{2+}$ concentrations (Figs. 2 F-J & 3 G-L) were several orders of magnitude higher at
shallower D-stations (max. 722 and 395 µM at NDT3-D and SDT3-D, respectively) compared to
depocenter stations (max. 13 and 51 µM at NDRO and SDRO, respectively). NDT3-C porewater
$Fe^{2+}$ concentration (Fig. 2G) peaked in the top 1 cm of sediment (similar to deeper stations)
while SDT3-C porewater $Fe^{2+}$ concentration (Fig. 3H) peaked around 5-cm sediment depth. $Fe^{2+}$
concentrations reached a max. at 0-2 cm and declined sharply with depth in depocenter and A-
station sediment. Northern basin sediment was similar, but the decline in $Fe^{2+}$ below 0-2 cm was
less pronounced.

Maximum porewater sulfide concentrations (Figs. 2 F-J & 3 G-L) were several orders of
magnitude lower at the shallower D-stations (5 and 4 µM at NDT3-D and SDT3-D, respectively)
compared to A stations (350 and 148 µM at NDT3-A and SDT1-A, respectively). Unlike $Fe^{2+}$,
peaks in sulfide concentration occurred deeper in the sediment (e.g., below 5 cm depth at A
stations). Porewater sulfate concentrations (Figs. 2 K-O & 3 M-R) decreased slightly with depth,
but never reached values below 20 mM at any station.

**3.4 In-situ microprofiling**
Microprofiler $O_2$ and sulfide measurements are shown in Fig. 4. Oxygen was rapidly consumed
within the first 0-1 cm of sediment at every station where $O_2$ was detected in the bottom water
(i.e., at all stations except NDRO, which showed no positive signal of oxygen in the water
compared to the sediment; note that no oxygen profile is available for SDRO). Sulfide
concentrations from microsensors showed similar trends to spectrophotometric measurements,
albeit with different absolute values (below detection in shallower B-, C- and D-stations that
lacked mats and >1,000 µM at A- and depocenter stations). Microprofiler pH (Fig. 4) was near
7.5 in the bottom water at all stations, and slowly decreased to near 7.0 in the lower parts (3-5
cm) sediment at most stations except NDT3-C and SDT3-B. pH at 2.5 cm at SDT3-B reached
6.77, which was the lowest observed during this expedition.

**Figure 2.** Biogeochemical data from ROV sediment push cores collected at stations on the northern transect
(NDT3) and in the northern depocenter (NDRO): total alkalinity (TA), dissolved inorganic carbon (DIC),
ammonium ($NH_4^+$), phosphate ($PO_4^{3-}$) in the first row; nitrate ($NO_3^-$), total sulfide (sulfide), and iron (II) ($Fe^{2+}$)
in the second row; sulfate ($SO_4^{2-}$) and bacterial sulfate reduction rate (SRR) in the third row. Data analyzed from
sediment core supernatant are plotted at -1 cm sediment depth; the dotted line connotes the sediment-water
interface. Note the change in scale on the primary x-axis in panel I and the change in scale of the secondary x-
axis in panels F and I. No spectrophotometric sulfide data is available for NDRO and NDT3-B and no SRR data
is available for NDT3-B. For station details see Fig. 1 and Table 1.

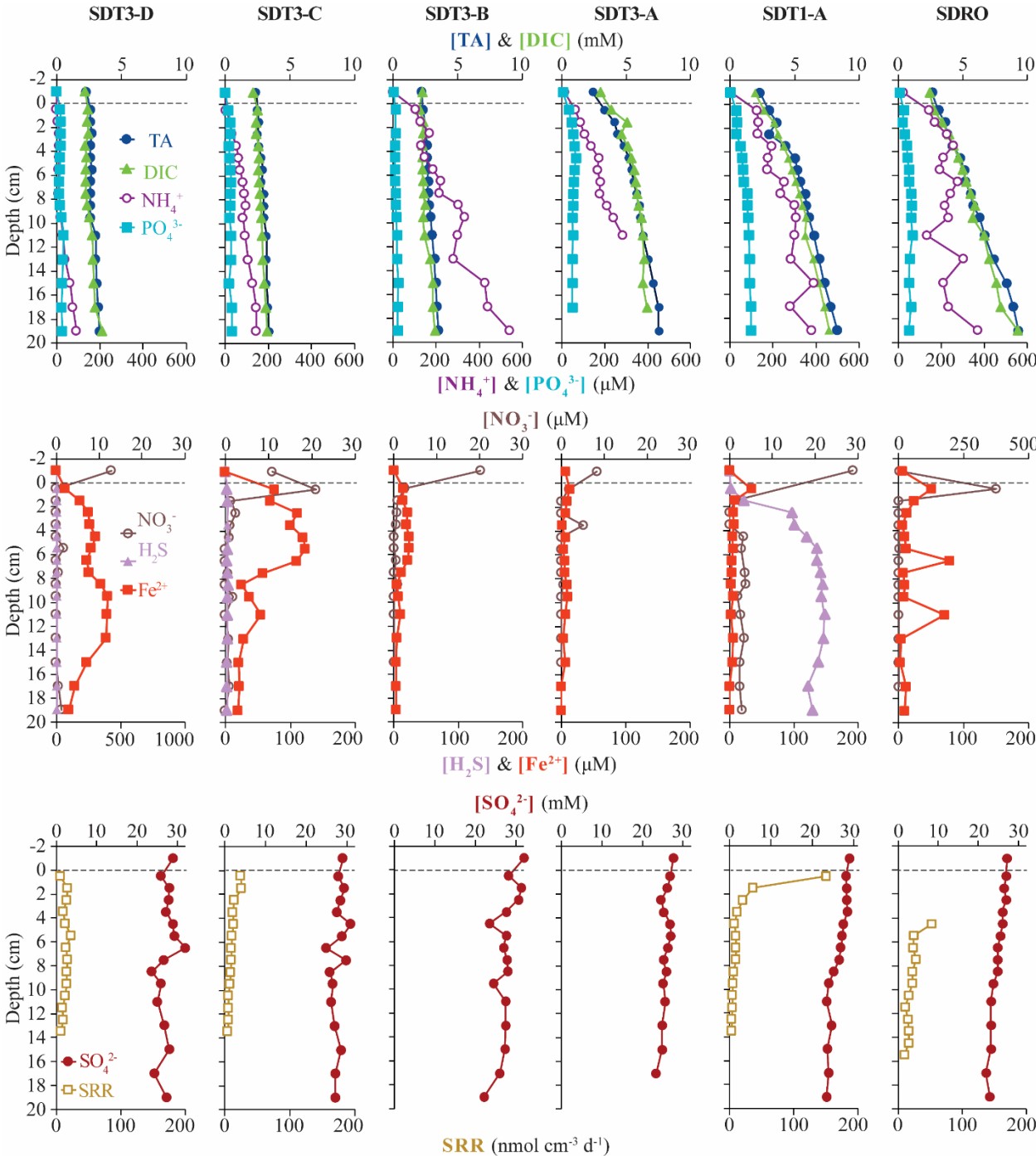

**Figure 3**. Biogeochemical data from ROV sediment push cores collected at stations on the two southern
transects (SDT1 and SDT3) and the southern depocenter (SDRO): total alkalinity (TA), dissolved inorganic
carbon (DIC), ammonium ($NH_4^+$), phosphate ($PO_4^{3-}$) in the first row; nitrate ($NO_3^-$), total sulfide (sulfide), and
iron (II) ($Fe^{2+}$) in the second row; sulfate ($SO_4^{2-}$) and bacterial sulfate reduction rate (SRR) in the third row. Data
analyzed from sediment core supernatant are plotted at -1 cm sediment depth; the dotted line connotes the
sediment-water interface. Note the change in scale on the primary x-axis in panel L and the change in scale of
the secondary x-axis in panel G. No sulfide nor SRR data are available for SDT3-B and -A; spectrophotometric
sulfide and the top 0-4 cm of SRR data are not available for SDRO. For station details see Fig. 1 and Table 1.


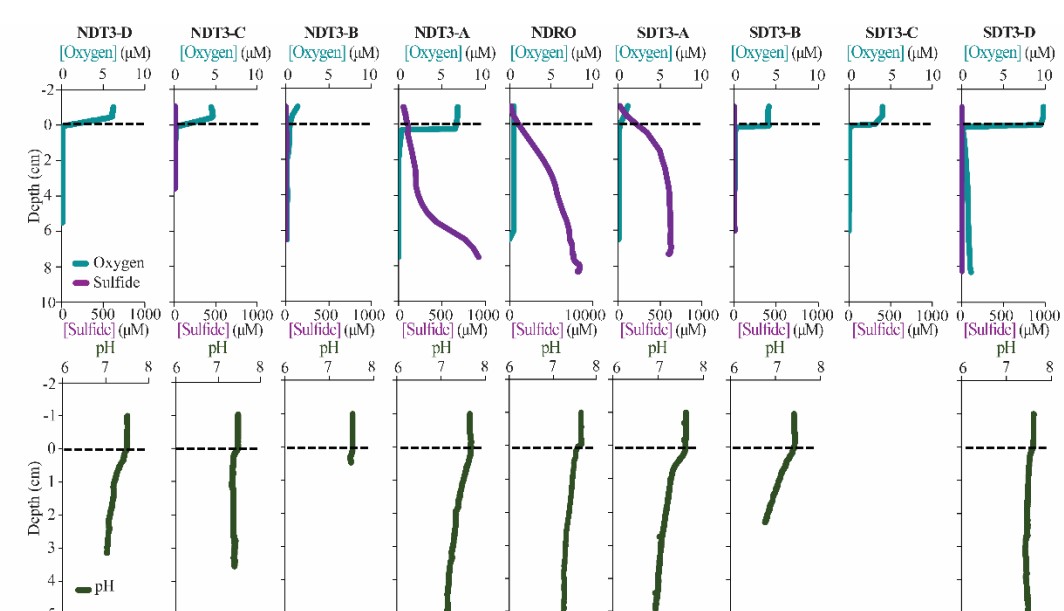

**Figure 4.** In-situ sediment microprofiler results for all stations (except SDT1-A and SDRO): oxygen ($O_2$) and
total sulfide (sulfide) concentration in the first row; pH profiles in the second row. Note the change in scale on
the secondary x-axis for NDRO sulfide. Values determined in the overlying water are plotted at negative
sediment depths; the dotted line connotes the sediment-water interface.

### 3.5 In-situ fluxes of benthic solutes

$NO_3^-$, $NH_4^+$, $PO_4^{3-}$, and $Fe^{2+}$ flux measured in the BFC revealed different patterns of uptake and

release from the sediment throughout the basin (Fig. 5 and Suppl. Figs. 1-4). BFC $O_2$

concentrations were compromised by $O_2$ release from the chamber's polycarbonate walls, which

prevented an accurate calculation of $O_2$ fluxes from BFC sensor data. $NO_3^-$ was consumed at all

stations as indicated by a negative flux (i.e., a flux into the sediment). On the contrary, benthic

release (i.e., a flux out of the sediment) was observed for all other analyzed solutes ($NH_4^+$, $PO_4^{3-}$,

and $Fe^{2+}$), with the lowest fluxes in the shallow D and C-stations and highest fluxes in the

depocenter. Ammonium fluxes were the highest of all the determined solutes and showed the

largest difference between deep and shallow stations, with a flux of 1.6 mmol $m^{-2}$ $d^{-1}$ at NDT3-C

(there were no measurable $NH_4^+$ fluxes in D-station chambers) and reaching $11.1 \pm 3.1$ mmol $m^{-2}$
$d^{-1}$ (n = 3) at the two depocenter stations. The depocenter ammonium flux far-outpaced the
concomitant flux of nitrate into depocenter sediments (-3.2 ± 0.7 mmol $m^{-2}$ $d^{-1}$, n = 3). Iron and
phosphate fluxes were similar at depocenter stations (4.1 ± 0.7, n = 3, and 3.2 ± 0.7, n = 3, mmol
$m^{-2}$ $d^{-1}$, respectively). Alkalinity and DIC concentrations from flux chambers (Suppl. Figs. 5 and
6) remained constant at all stations and thus no DIC flux was calculated. Results from BFCs
injected with $^{15}N$-$NO_3^-$ at the SDRO and NDT3-D station are shown in Fig. 6. The rates of
denitrification, anammox, and $N_2O$ production were higher at SDRO compared to NDT3-D.
$^{15}NH_4^+$ production (DNRA) was one order of magnitude higher at the SDRO station (2.67 mmol
$m^{-2}$ $d^{-1}$) compared to the NDT3-D station (0.14 mmol $m^{-2}$ $d^{-1}$). DNRA accounted for a much
higher percentage of $NO_3^-$ reduction at SDRO (54.1%) than NDT3-D (13.3%).

**3.6 Sulfate reduction rates**
Vertical profiles of bacterial sulfate reduction as determined by the radioisotope technique
differed throughout the basin (Figs. 2 & 3). Peaks in sulfate reduction were seen in the top 0-1
cm of sediment at stations with a visible GSOB mat on the surface (120.2, 151.0, and 85.3 nmol
$cm^{-3}$ $d^{-1}$ at NDRO, SDT1-A, and NDT3-A, respectively). Sediments at most shallower basin
depths exhibited peaks slightly deeper in the sediment and of lower magnitude (25.5, 44.5, 22.5
nmol $cm^{-3}$ $d^{-1}$ at SDT3-C, NDT3-D, and SDT3-D respectively). NDT3-C had no visible GSOB
mats but exhibited a peak (133.7 nmol $cm^{-3}$ $d^{-1}$) in sulfate reduction at 0-1 cm depth, similar to
deeper stations (e.g., NDRO in Fig. 2O), which differed from other shallow stations (e.g., SDT3-
C in Fig. 3N). The integrated sulfate reduction rate (0-14 cm depth) at NDRO (4.1 mmol $m^{-2}$ $d^{-1}$)
was noticeably higher than most other stations with the exception of NDT3-C (3.8 mmol $m^{-2}$ $d^{-1}$)
(Table 2). NDT3-D and NDT3-C exhibited higher integrated rates (2.9 and 3.8 mmol m$^{-2}$ d$^{-1}$)
than their southern station counterparts SDT3-D and SDT3-C (1.9 and 1.7 mmol m$^{-2}$ d$^{-1}$).

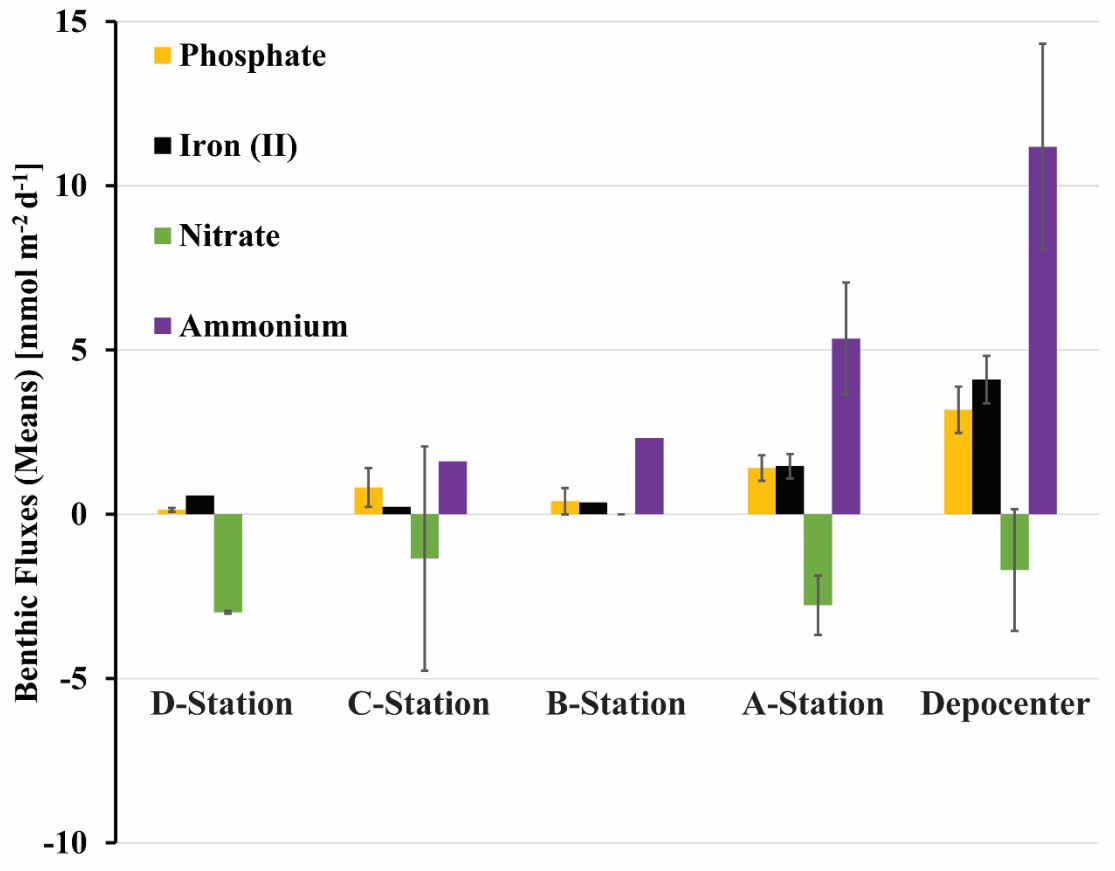

**Figure 5.** Benthic fluxes of solutes (positive flux = release from the seafloor; negative flux = uptake by the
seafloor) determined with in-situ benthic flux chambers. Rates were averaged for stations of same depth from
the northern and southern transect and the depocenter (NDRO and SDRO). Note, giant sulfur-oxidizing bacterial
mats were found at depocenter and A-stations. Error bars represent standard errors.

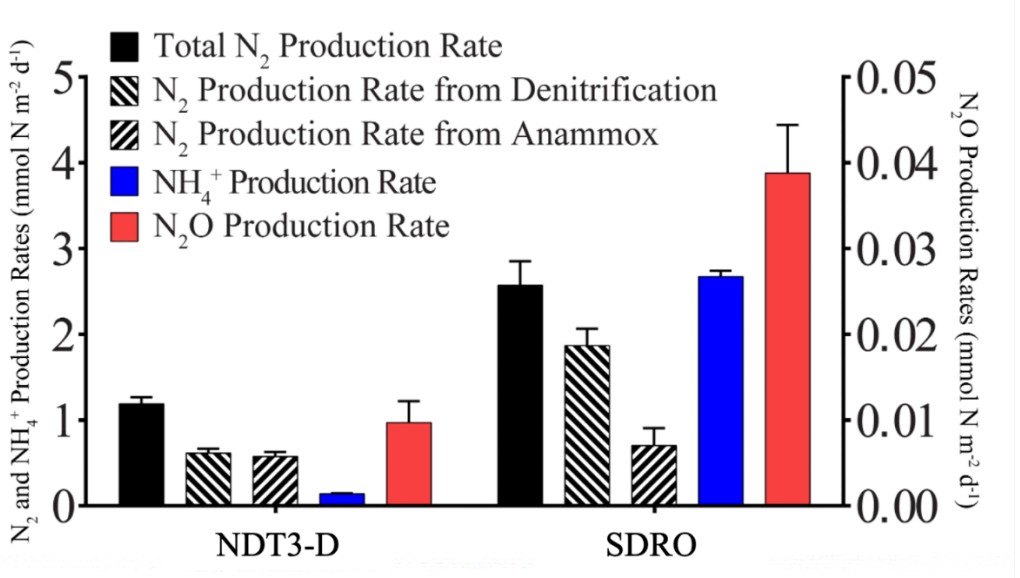

**Figure 6.** Areal rates of total $N_2$ production, denitrification, anammox, $NH_4^+$ production (DNRA), and $N_2O$
production


## 4. Discussion

### 4.1 Giant sulfur-oxidizing bacterial mats proliferated in response to deoxygenation in the Santa Barbara Basin

The SBB is an ideal environment to study the effect of transient deoxygenation on benthic biogeochemistry. In Fall 2019, when this expedition took place, the SBB was undergoing a transition from oxygenated to virtually anoxic conditions (Qin et al., 2022). When the AT42-19 cruise occurred, most of the bottom water in the basin was hypoxic (A-, B-, C-, and D-stations), except for the depositional center. Separate $O_2$ measurements from the ROV sensor ($O_2$ below detection limit, Table 1), microprofilers (no signal change between water column and sediment, Fig. 4), and Winkler titration from CTD/rosette casts (uniform non-zero value below 500 m (Qin et al., 2022)) indicated full anoxia in the bottom water at the deeper stations (NDRO and SDRO). Notably, bottom water conditions revealed a slight asymmetry between the basin transects (Fig. 1); bottom water along the northern transect generally had more $O_2$ and $NO_3^-$ than the southern transect (e.g., 9 μM $O_2$ at NDT3-A and 0 μM $O_2$ at SDT3-A). This asymmetry indicated differences in the circulation and/or microbial communities between the northern and southern portions of the basin. Whether this asymmetry is a permanent feature of the basin or symptomatic of the specific conditions in November 2019 is unclear; previous studies in the SBB have been restricted to the depocenter or one side of the basin (Sholkovitz, 1973; Reimers et al., 1996; Kuwabara et al., 1999). Regardless of bottom water oxidant concentration, the energetically most favorable terminal electron acceptors ($O_2$ and $NO_3^-$) disappeared in a very narrow zone below the sediment-water interface (Fig. 4 and Figs. 2 and 3, respectively), consistent with their expected rapid consumption by the benthic microbial community.

In the present study, benthic GSOB mats were primarily limited to the anoxic depocenter of the
SBB. Similarly, such mats were replete in the core of the anoxic Peruvian OMZ (Levin et al.,
2002; Sommer et al., 2016; Mosch et al., 2012), but absent from the seafloor below the hypoxic,
i.e., slightly oxygenated, Mauritanian OMZ (Schroller-Lomnitz et al., 2019). GSOB mats in
November 2019 were observed deeper in the basin than in October 2013 (Valentine et al., 2016)
but in a similar location to June 1988  (Reimers et al., 1996) and April 1997 (Kuwabara et al.,
1999). During the 2013 sampling, dense GSOB mats were confined to depths between approx.
500-570 m (equivalent to the B-stations from this expedition), corresponding with anoxic
conditions in the bottom water. This habitat was sandwiched between an anoxic, anitric (i.e.,
nitrate-free) deep and a hypoxic, nitrigenated (i.e., nitrate-rich) shallower water layer (Valentine
et al., 2016). The difference in depth distribution of GSOB mats between the 2013 and 2019
expedition provides evidence that GSOB mats in the SBB are ephemeral and proliferate where
the bottom water is anoxic but not anitric.

As our study represents only a snapshot of an oxygen- and nitrate-driven mat dynamic, we can
only speculate how areas of the basin that did not contain GSOB mats in November 2019 fit into
this dynamic. For example, mat-forming sulfur bacteria found slightly deeper in the sediment at
B-stations (see section 3.2) could be progenitors to surface sediment colonization of thick GSOB
mats, as has been recorded in other transiently deoxygenated environments (Jørgensen, 1977).
Alternatively, these subsurface colonies could also be remnants of a former surface GSOB mat
that retreated under changing redox conditions. Oxygenated conditions in the water preceding
the 2019 expedition would, in this context, suggest the mats migrated following a previous
anoxic event (Qin et al., 2022). If deoxygenation persisted in the SBB after the AT42-19 cruise,
then anitria (i.e., anitric conditions – similar to anoxia) would likely follow in the deepest basin
water. These conditions would be similar to those seen in 2013 (Valentine et al., 2016), where
GSOB mats formed a contiguous "donut ring" at shallower basin depths. Interestingly, GSOB
mats in the Eastern Gotland Basin of the Baltic Sea were confined to a hypoxic transition zone,
where $O_2$ was < 30 µM but did not reach anoxia, while no mats were observed at deeper anoxic
locations (Noffke et al., 2016). This difference in distribution compared to the SBB suggests that
GSOB mats proliferate under different conditions (anoxic or hypoxic), potentially depending on
the species of mat-forming bacteria present and whether they specialize in aerobic or anaerobic
chemosynthesis.

**4.2 Shift from benthic denitrification to dissimilatory nitrate reduction to ammonium in**
**response to complete deoxygenation in the Santa Barbara Basin**

Benthic uptake and release of nitrogen species by SBB sediment appeared to be affected by the
presence of GSOB mats. While total benthic nitrate uptake was similar between D- and
depocenter stations based on in-situ $NO_3^-$ flux measurements (Fig. 4), $NH_4^+$ release from the
sediment into the water column increased where GSOB mats were present (Fig. 5). This trend is
supported by the porewater profiles of $NH_4^+$, which showed a steeper increase over sediment
depth at deeper stations (Figs. 2 & 3). Incubations with $^{15}N$-$NO_3^-$ revealed that $N_2$ production
(denitrification and anammox) accounted for 86% of $NO_3^-$/$NO_2^-$ reduction in the shallow basin,
while $NH_4^+$ production (DNRA) accounted for 13% and $N_2O$ production accounted for 1%
(NDT3-D, Fig. 6;(Peng et al., 2023)). In contrast, most (54%) of $NO_3^-$ reduction at the
depositional center occurred via DNRA; $N_2$ production accounted for 45% and $N_2O$ production
accounted for 1% of $NO_3^-$ reduction at the SDRO (Fig. 6; Peng et al. 2023). It is important to
note that these results only describe patterns of $NO_3^-$ reduction in the basin, while other
mechanisms of nitrate uptake by sediment (e.g., hyper-accumulation of nitrate into vacuoles) are
more difficult to calculate accurately.

It is likely important to SBB benthic nitrogen cycling that some eukaryotic organisms can hyper-
accumulate $NO_3^-$ in benthic, anoxic environments including diatoms (Kamp et al., 2011) and
foraminifera (Risgaard-Petersen et al., 2006). Additionally, meiofauna (e.g., nematodes) can
enhance rates of denitrification  (Bonaglia et al., 2014). Both foraminifera and meiofauna were
observed in SBB depocenter and A-station sediments in November 2019, and diatoms were
observed in shallower sediments in the basin (data not shown). Other studies found that benthic
foraminifera in the SBB depocenter can hyperaccumulate $NO_3^-$ intracellularly up to $375 \pm 174$
mM (Bernhard et al., 2012) and host symbionts capable of performing denitrification (Bernhard
et al., 2000). These foraminifera were found to be responsible for approx. 3 mM N $m^{-2}$ $d^{-1}$, or
67% of the total denitrification occurring in the SBB depocenter (Bernhard et al., 2012).
Additionally, fungi could reduce $NO_3^-$ or $NO_2^-$ to nitrous oxide in marine sediments and may
contribute to denitrification in SBB sediments (Kamp et al., 2015; Lazo-Murphy et al., 2022).
This opens up the possibility that the majority of denitrification we observed in the SBB
depocenter is performed by eukaryotes, while prokaryotes (especially GSOB) are responsible for
most of the DNRA. Elevated $NO_3^-$ and $NO_2^-$ concentrations that were observed in our 0-1 cm
samples from NDT3-A and SDRO have been reported from SBB depocenter sediments in the
past (Reimers et al., 1996; Bernhard et al., 2003), and have been attributed to both GSOB and benthic
eukaryotes. The impact of eukaryotes on SBB benthic nitrogen transformation remains to be disentangled
from the mats themselves.


Our data suggests a transition from denitrification-dominated sediment in the oxygenated basin
to an increasing influence of DNRA on N cycling in the deeper, anoxic basin. Placed in the
context of other OMZs, Mauritanian shelf sediment was dominated by denitrification (Dale et al.,
2014), similar to SBB shallow sediment (below hypoxic water) while core Peruvian OMZ
sediment was dominated by DNRA, similar to sediment of the deeper SBB (below anoxic water)
(Sommer et al., 2016). Nitrate reduction in sediment below the seasonally hypoxic Eckernförde
Bay (Dale et al., 2011) and below the hypoxic transition zone of the Eastern Gotland Basin
(Noffke et al., 2016) also showed increased DNRA where GSOB mats were present, though with
an order of magnitude lower $NH_4^+$ flux (avg. 1.74 mmol $m^{-2}$ $d^{-1}$ and max. 1.10 mmol $m^{-2}$ $d^{-1}$,
respectively) than the SBB depocenter.

While our study suggests a shift from denitrification to DNRA during deoxygenation of SBB
bottom water, other studies examined changes in benthic nitrogen cycling under reverse
conditions, i.e., the reoxidation of the environment following anoxia  (Hylén et al., 2022; De
Brabandere et al., 2015). After a decadal oxygenation event in the Eastern Gotland Basin (Baltic
Sea) in 2015-2016, sediment exhibited a slight increase in denitrification, but remained
dominated by DNRA and $N_2O$ production (Hylén et al., 2022). The lack of $N_2$ production via
denitrification following this oxygenation event was attributed to the reoxygenation event being
too weak to substantially oxidize sediments, which would favor denitrification (Hylén et al.,
2022). In an engineered reoxygenation event of the By Fjord on Sweden's western coast, where
dissolved $O_2$ and $NO_3^-$ content of anoxic and anitric bottom water was artificially increased to
approx. 130 µM $O_2$ and 20 µM $NO_3^-$ over a period of roughly 2 years, denitrification rates were
increased by an order of magnitude and DNRA rates were also stimulated (De Brabandere et al.,
2015). Comparing our results to these two studies suggests that DNRA bacteria are more
resilient to weak reoxygenation events and thrive in transiently deoxygenated systems that
remain hypoxic ($O_2 < 63$ µM). The frequency and magnitude of reoxygenation and
deoxygenation of SBB bottom waters, and the effect of these processes on the benthic microbial
community, could be a major factor supporting some of the highest recorded total nitrate
reduction rates in a natural benthic marine setting (Peng et al., 2023).
**Table 3.** Example reactions of nitrate reduction pathways with associated energy yield in respect to the electron
donor ($H_2$ or $HS^-$) and electron acceptor ($NO_3^-$) and electron accepting capacity. Modified from Table 2 in (Tiedje
et al., 1983).

| Reaction | $\Delta G^{o'}$ (kcal mol$^{-1}$) | | Electrons per $NO_3^-$ |
|---|---|---|---|
| | $H_2$ / $HS^-$ | $NO_3^-$ | |
| Chemoheterotrophic Denitrification | | | |
| $2NO_3^- + 5H_2 + 2H^+ \rightarrow N_2 + 6H_2O$ | -53.6 | -133.9 | 5 |
| Chemoautotrophic Denitrification | | | |
| $8NO_3^- + 5HS^- + 3H^+ \rightarrow 5SO_4^{2-} + 4N_2 + 4H_2O$ | -177.9 | -111.2 | 5 |
| Chemoheterotrophic DNRA | | | |
| $NO_3^- + 4H_2 + 2H^+ \rightarrow NH_4^+ + 3H_2O$ | -35.8 | -143.3 | 8 |
| Chemoautotrophic DNRA | | | |
| $NO_3^- + HS^- + H^+ + H_2O \rightarrow NH_4^+ + SO_4^{2-}$ | -107.0 | -107.0 | 8 |



A high ratio of electron donor to electron acceptor favors DNRA over denitrification  (Marchant
et al., 2014; Hardison et al., 2015; Tiedje et al., 1983) and this ratio appears to be critical in
determining the dominant nitrate reduction pathway in SBB sediments, similar to the Eastern
Gotland Basin (Hylén et al., 2022) and the By Fjord (De Brabandere et al., 2015). Example
energy yields for denitrification and DNRA are shown in Table 3.  As discussed in (Tiedje et al.,
1983), heterotrophic denitrification yields more energy per mol of electron donor than DNRA.
However, the reverse is true when considering energy yield per mol of electron acceptor ($NO_3^-$).
DNRA also yields 3 more electrons per molecule of $NO_3^-$ than denitrification. Tiedje et al.
argued that in environments that are starved of powerful terminal electron acceptors, such as
anoxic, organic-rich sediment, the energy yield per electron acceptor and additional electrons
available for transfer could push nitrate reduction towards DNRA. Multiple laboratory and
model studies have converged on an electron donor to acceptor ratio of approximately 3 to
encourage DNRA over denitrification (Hardison et al., 2015; Algar and Vallino, 2014) though
other studies have found higher values (Porubsky et al., 2009; Kraft et al., 2014). Sulfide
concentrations near the sediment-water interface at the SBB depocenter (approx. 200 µM at 0.5
cm depth; Fig. 3, NDRO) would favor chemoautotrophic DNRA over denitrification at ambient
marine nitrate concentrations (approx. 28 µM). Additionally, DNRA appears to be the preferred
nitrate reduction pathway for chemoautotrophs that utilize iron or sulfide as an electron donor
(Caffrey et al., 2019; Kessler et al., 2019; An and Gardner, 2002). As GSOB mats hyper-
accumulate nitrate from the bottom water into their intracellular vacuoles, the resulting decline in
electron acceptors at the sediment-water interface coupled with an elevation of the sulfate
reduction zone would create an electron donor to acceptor ratio that favors DNRA. Since GSOB
mats in the SBB seem to prefer DNRA, starving the bottom water of electron acceptors coupled
with the high sulfate reduction rates could give them a competitive advantage and allow them to
proliferate into the largest-yet mapped GSOB mat in Earth's oceans, as seen in other expeditions
(Valentine et al., 2016; Reimers et al., 1996; Kuwabara et al., 1999).

**4.3 Microbial mat proliferation and benthic phosphate remineralization dependent on high**
**rates of organic matter degradation in the Santa Barbara Basin**

Organic carbon content of the benthic environment appears to be a key control on sulfate
reduction rates near the sediment-water interface as well as microbial mat proliferation. Sulfate
reduction rates in the SBB depocenter are most similar in magnitude and profile (i.e., highest
rates found at the sediment-water interface and decline drastically thereafter) to those found in
sediments below the transiently deoxygenated portion of the Peruvian shelf (e.g., 4.1 mmol m$^{-2}$
d$^{-1}$ at the SBB NDRO station vs. 2.5-3.8 mmol m$^{-2}$ d$^{-1}$ at 128-144 m water depth on the Peruvian
margin (Gier et al., 2016; Treude et al., 2021)). The TOC content of surface sediments in these
two regions are both high and within the same order of magnitude (maximum recorded TOC of
5.2% at the 0-1 cm margin at the SDT1-A station compared with 7.6% in the Peruvian margin
145 m depth (Noffke et al., 2012)). In comparison, sulfate reduction rates in the SBB were at
least one order of magnitude lower than found in sediment below the OMZ on the Namibian
Shelf, which has much higher TOC contents of >10% (Brüchert et al., 2003; Bremner, 1981).
Sulfate reduction rates in the shelf sediments below the Eastern Arabian OMZ were much lower
(0.18 – 1.27 mmol m$^{-2}$ d$^{-1}$) than rates in the SBB depocenter (Naik et al., 2017) despite similar
hypoxic to anoxic bottom water conditions. These lower sulfate reduction rates were attributed to
the relatively low amount of pelagic primary productivity and ergo benthic organic matter
delivery in the Eastern Arabian OMZ compared to other upwelling systems (Naik et al., 2017).
The organic matter content of the sediment appears to be important in the proliferation of GSOB
mats; too much TOC could result in toxic levels of sulfide at the sediment-water interface
(*Beggiatoa* exhibit an aversion to sulfidic sediments but toxicity has not been quantified)
(Preisler et al., 2007), whereas too little sulfide would not provide enough electron donor for the
GSOB's chemoautotrophic metabolism.

The profiles of several indicators for benthic anaerobic organic matter remineralization (total
alkalinity, DIC, $PO_4^{3-}$, $NH_4^+$) increased in steepness with increasing water depth (Figs. 2 A-E &
3A-F). One divergence from this trend can be seen in $PO_4^{3-}$ profiles from the shallow C- and D-
stations, which also featured low rates of sulfate reduction. $PO_4^{3-}$ profiles in these sediments
track closely to $Fe^{2+}$ profiles; both solutes dip in concentration in areas with visible iron sulfide
formation (e.g., 5-11 cm in NDT3-D as seen in Fig. 2A). Additionally, several stations that
exhibited high sulfate reduction rates in surface sediment (e.g., SDT1-A) showed almost no
change in $PO_4^{3-}$ at depths below 5 cm (e.g., Fig. 2 K-O compared to Fig. 2 A-E). This
phenomenon has been previously documented in SBB sediment and is attributed to the
precipitation of carbonate fluorapatite (Reimers et al., 1996). The confinement of these flat $PO_4^{3-}$
profiles to stations with >100 nmol $cm^{-3}$ $d^{-1}$ sulfate reduction in surface sediment suggests that
this mineralogical sink of $PO_4^{3-}$ in SBB sediment may be dependent on high sulfate reduction
rates, owing to the bicarbonate produced by sulfate reduction (Reimers et al., 1996), and is not
found throughout the basin. Flat $PO_4^{3-}$ profiles were also reported from the transiently
deoxygenated portion of the Peruvian OMZ, where phosphate mineral precipitation has been
documented (Noffke et al., 2012). Similar to the shallow margins of the SBB, $PO_4^{3-}$ in
Mauritanian OMZ porewater tracks closely with changes in porewater $Fe^{2+}$ (Schroller-Lomnitz
et al., 2019), indicating that iron mineralization/dissolution mechanisms hold a greater influence
on $PO_4^{3-}$ concentrations under hypoxic bottom waters.

**4.4 Iron oxide exhaustion is critical for raising the sulfate reduction zone close to the**
**sediment-water interface in Santa Barbara Basin sediment.**

The hyper-accumulation of $NO_3^-$ by GSOB mats potentially facilitates sulfate reduction close to
the sediment-water interface in the SBB (e.g., NDRO and NDT3-A as seen in fig. 2N and 2O) by
starving the sediment of this more powerful electron acceptor. The rise of the sulfate reduction
zone at NDT3-C (fig. 2L) further suggests that the exhaustion of iron oxides and the formation of
iron sulfide below the sediment-water interface may play a crucial role in controlling the
distribution of sulfate reduction as well. SBB sediments showed a wide vertical and horizontal
heterogeneity of redox states based on visual appearance (Fig. 1A-K). Sediment beneath the
hypoxic bottom water at the shallowest D-stations was reddish, consistent with a high content of
iron oxides. Interestingly, porewater $Fe^{2+}$ concentrations in shallower parts of the basin (e.g.,
NDT3-D, max. ~700 μM $Fe^{2+}$) were an order of magnitude larger than those found in both the
Peruvian (max ~60 and ~30 μM $Fe^{2+}$, respectively; (Noffke et al., 2012; Plass et al., 2020) and
Mauritanian (max. ~50 μM $Fe^{2+}$; Schroller-Lomnitz et al 2019) OMZ. It should be noted that
porewater samples for geochemical analyses were unfiltered and hence reported iron
concentrations include aqueous, colloidal, and nanoparticulate species. Regardless, all these
components represent bioavailable sources of iron. Further, since filtering through 0.45 or 0.2
μm filters only removes a fraction of colloidal particles and no nanoparticles (Raiswell and
Canfield, 2012), potential surplus porewater iron in SBB samples in comparison to studies that
applied filtering was likely minimal.

Deeper in the basin, bands of black sediment that appear mid-core at NDT3-C (6-14 cm) and
SDT3-C (6-10 cm) indicate the formation of iron sulfides as a result of sulfide produced by
sulfate reduction (Canfield, 1989). Both D-stations had similar bottom water conditions (Table
1), sulfate reduction rates (Fig. 3W-AG), porewater concentrations of solutes (Figs. 2 and 3), and

732 visual sediment characteristics (Section 3.1). On the contrary, there are some noticeable

733 differences in the porewater geochemistry between the two C-stations. At the C-stations, peaks

734 in sulfate reduction were in the surface sediment, above the iron sulfide layers, and declined

735 below approximately 4 cm, indicating a discrepancy between observed peak sulfate reduction

736 activity and the mineralogical clues left behind by the process.  Comparing NDT3-C and SDT3-

737 C, iron sulfide formation (Table 1B compared to 1J), porewater $Fe^{2+}$ profiles (Fig. 2G compared

738 to Fig. 3H), and sulfate reduction rates (Fig. 2L compared to Fig. 3N) show that NDT3-C

739 sediment appears to be in transition towards a more sulfidic state, while SDT3-C sediments still

740 mimic the shallow D-station ferruginous state. While sulfate reduction rates for B-stations are

741 not available due to technical issues during sample processing, porewater $Fe^{2+}$ profiles show a

742 similar difference between the north and south basin (Fig. 2H compared to Fig. 3I) as did visual

743 sediment characteristics (Table 1C compared to 1I). This difference in biogeochemical profiles

744 and apparent minerology between the north and south C- and B-stations could be a result of

745 hydrographic and/or bathymetric differences in the basin (Sholkovitz and Gieskes, 1971; Bograd

746 et al., 2002), but a discernable link between the differences in sediment biogeochemistry and the

747 differences in bottom water oxygen (Table 1) need to be further explored.

749 Deeper in the basin (depocenter and A-stations), porewater $Fe^{2+}$ concentrations in sediment

750 beneath anoxic bottom water (max. 84 µM $Fe^{2+}$) were similar to concentrations found below the

751 Peruvian OMZ in 2008 under anoxic bottom water conditions (78 m water depth, max. 80 µM

752 $Fe^{2+}$) (Noffke et al., 2012). These deep basin porewater $Fe^{2+}$ concentrations were, however, an

753 order of magnitude larger than those found at a similar site on the Peruvian shelf (75 m water

754 depth, max. 1 µM $Fe^{2+}$) in 2017 during a kelvin-wave-associated "Coastal El Niño" event that

created oxygenated bottom waters during the sampling and the disappearance of previously
observed dense GSOB mats (Plass et al., 2020). As the SBB water column was undergoing rapid
deoxygenation in the weeks preceding this study (Qin et al., 2022), the sediments below the sill
appeared to be actively shifting from a ferruginous state to a sulfidic state, with this change
starting around the C-stations and being complete in the depocenter. Comparing apparent iron
sulfide formation with dips in porewater $Fe^{2+}$ concentrations in C-station profiles (Fig. 1B
compared to Fig. 2G and Fig. 1J compared to Fig. 3H) signals a shift away from a ferruginous
state occurring just below the SBB sill.

C-station porewater $Fe^{2+}$ concentrations and sulfate reduction rates indicate that migration of the
sulfate reduction zone towards the sediment-water interface is associated with iron sulfide
formation deeper in the sediment. The activity (or lack thereof) of cable bacteria, which are able
to bridge the gap between the oxidized sediment-water interface and reduced sediment below
using a biofilament (Pfeffer et al., 2012), could explain the interplay between sulfate reduction
and iron cycling in SBB sediments. Cable bacteria, such as *Ca. Electronema* sp., contain  genes
involved in DNRA (Kjeldsen et al., 2019) and can perform nitrate reduction in incubation
experiments (Marzocchi et al., 2014), but their direct transformation of $NO_3^-$ in the environment
appears limited (Kessler et al., 2019) and they appear to be inactive in anoxic aquatic
environments (Seitaj et al., 2015; Marzocchi et al., 2018; Hermans et al., 2019). Cable bacteria
primarily conduct aerobic sulfide oxidation (Pfeffer et al., 2012), though they can also utilize
$Fe^{2+}$ as an electron donor (Seitaj et al., 2015). The maximum recorded filament length of cable
bacteria is 7 cm (Van De Velde et al., 2016), though typically they are not stretched completely
vertically through the sediment. The appearance of black sediment in the SBB C-station
sediments, starting at approx. 5 cm depth, could be an indication that cable bacteria are oxidizing
iron sulfides at that sediment depth and prevent their formation at shallower depths. Further,
cable bacteria have been found to directly compete with GSOB in transiently deoxygenated
systems, with cable bacteria active under oxygenated conditions and GSOB active in anoxic
conditions (Seitaj et al., 2015).  Cable bacteria can also prevent the benthic release of sulfide,
which is toxic to many pelagic animals, via the creation of an iron-oxide buffer (formed through
$Fe^{2+}$ oxidation) in near-surface sediments (Seitaj et al., 2015). Therefore, if cable bacteria activity
in the SBB decreased with declining oxygen concentrations below the sill, the iron oxide buffer
they create could have been reduced, encouraging the sulfate reduction zone to migrate towards
the sediment surface (as seen at NDT3-C). Cable bacteria can sometimes be detected in
sediments via a slight pH increase (typically pH > 8) (Schauer et al., 2014) which was not
reflected in our pH results, but this phenomenon is more typically seen in the laboratory and not
the field (Hermans et al., 2019).

**4.5 Iron and phosphate flux into SBB bottom water is a feature of transient deoxygenation.**

Release of dissolved iron and phosphate from sediment below anoxic waters is a well-
documented phenomenon (e.g., (Mortimer, 1941; Van Cappellen and Ingall, 1994; Van De
Velde et al., 2020; Noffke et al., 2012)) and this phenomenon is seen in the SBB as well. As
postulated previously (Kuwabara et al., 1999), basin flushing oxidizes iron sulfides at the
sediment-water interface, providing ample substrate for microbial iron reduction once anoxia
returns. This iron reduction initiates high rates of $Fe^{2+}$ release from SBB depocenter sediment
(Fig. 5). Iron reduction further releases iron-bound $PO_4^{3-}$ (Mortimer, 1941) as seen by high
benthic fluxes of $PO_4^{3-}$ at the depocenter (Fig. 5), although notably some of this $PO_4^{3-}$ release is
likely attributed to organic matter degradation (Van Cappellen and Ingall, 1994). High benthic
$Fe^{2+}$ and $PO_4^{3-}$ fluxes were also seen on the Peruvian shelf during transient anoxia (Noffke et al.,
2012). The release of these solutes was interpreted to be sourced from a layer of reactive iron
hydroxides existing near the sediment surface, likely established during a recent oxygenation
event. Similar conditions, i.e., visibly oxidized (reddish) sediment laminae and a thin zone of
iron reduction apparent from a peak in $Fe^{2+}$ at the sediment-water interface, were found in
sediment from the SBB depocenter. Deeper in the persistently anoxic core of the Peruvian OMZ,
sediment appears to have little to no flux of $Fe^{2+}$ and $PO_4^{3-}$ into the bottom water (Noffke et al.,
2012). Here, iron at the sediment-water interface is hypothesized to be locked up in iron sulfides,
which are rarely re-oxidized due to persistent anoxia.

In a different study from the Eastern Gotland Basin in the Baltic Sea, enhanced elemental fluxes
were observed during a decadal oxygen flushing event (Van De Velde et al., 2020), which was
attributed to enhanced elemental recycling, or cycles of mineral precipitation in the water column
followed by mineral dissolution once those minerals sink to the sediment. Notably, the iron flux
observed in the Eastern Gotland Basin (max. 0.08 mmol $m^{-2}$ $d^{-1}$) (Van De Velde et al., 2020) was
two orders of magnitude lower than the flux observed in the anoxic depocenter of the Santa
Barbara Basin (max. 4.9 mmol $m^{-2}$ $d^{-1}$). It is further notable that benthic fluxes of $PO_4^{3-}$ in the
SBB depocenter were also an order of magnitude higher than fluxes in the Eastern Gotland
Basin's hypoxic transition zone (3.6 vs. 0.23 mmol $PO_4^{3-}$ $m^{-2}$ $d^{-1}$) - both of which contained
GSOB mats, but while the SBB was anoxic and the Eastern Gotland Basin was hypoxic (Noffke
et al., 2016). These differences in $Fe^{2+}$ and $PO_4^{3-}$ flux between the SBB and the Eastern Gotland
Basin suggest that reoxidation of the sediment-water interface during basin flushing, as opposed
to water-column-associated reoxidation, appears to encourage higher benthic iron fluxes.

$Fe^{2+}$ and $PO_4^{3-}$ flux from the SBB depocenter were also approximately five times higher (Fig. 5)
compared to the anoxic Peruvian shelf (4.9 vs. 0.9 mmol $Fe^{2+}$ $m^{-2}$ $d^{-1}$ and 3.6 vs. 0.8 mmol $PO_4^{3-}$
$m^{-2}$ $d^{-1}$, respectively) (Noffke et al., 2012). Based on $Fe^{2+}$ profiles, the zone of iron reduction in
Peruvian shelf sediments extended down to approx. 10 cm, while the zone appeared to be much
shallower and narrower (less than the top 5 cm) in the SBB depocenter. These differences in
magnitude of $Fe^{2+}$ concentration and $Fe^{2+}$ and $PO_4^{3-}$ flux between the SBB depocenter and the
Peruvian shelf could be attributed to differences in the recency and magnitude of reoxygenation
events. The release of $Fe^{2+}$ from sediment into the bottom water could create a buffer against
reoxygenation in transiently deoxygenated systems, giving a competitive advantage to anaerobic
benthic metabolisms (Dale et al., 2013; Wallmann et al., 2022). Additionally, both $Fe^{2+}$ and
$PO_4^{3-}$ release from the SBB sediment could allow for higher rates of primary productivity if
those constituents diffused into the photic zone (Robinson et al., 2022). The fate of $Fe^{2+}$ and
$PO_4^{3-}$ diffusing into SBB waters from the sediment-water interface is a focus of ongoing work
within the basin.
**5 Conclusions**

This research expands upon the wealth of science already conducted in the SBB and other
transiently deoxygenated environments by examining changes in benthic biogeochemistry
promoted by the onset of anoxia. Our main interpretations are summarized in Fig. 7. We found
that GSOB mats proliferate in the SBB where the bottom water is anoxic and nitrate
concentrations are declining (Fig. 7, A- and depocenter stations). Nitrate uptake by SBB
sediment is similar regardless of GSOB mat presence, but these mats appear to initiate a shift
from denitrification to DNRA as the primary nitrate reduction pathway (Fig. 7, beginning at B-
stations). The zone of sulfate reduction rises to the sediment-water interface where GSOB mats
are present (Fig. 7, A-stations), possibly because the hyper-accumulation of nitrate into their
intracellular vacuoles starves the environment of this more powerful electron acceptor. However,
following the natural order of electron acceptor utilization (Boudreau and Jorgensen, 2001), iron
oxides near the sediment-water interface must be exhausted before sulfate reduction can
dominate surface sediments and GSOB mats can proliferate in the SBB (Fig. 7, depocenter
stations). If anoxic events become longer and more frequent in the SBB because of global
warming (see, e.g., (Qin et al., 2022; Stramma et al., 2008)), the iron oxide buffer built up in
shallower basin depths could be exhausted, allowing for surface sulfate reduction and the
proliferation of GSOB mats in shallower margins of the basin than currently seen. Further, the
same transient deoxygenation that allows for these mats to re-stablish themselves also allows for
a high $Fe^{2+}$ and $PO_4^{3-}$ flux into the SBB water column. In order to fully understand the complex
changes in the benthic environment in response to deoxygenation, genomic and molecular work
of the upper sediment community needs to be characterized. Overall, the insights gleaned from
this research will aid in the understanding of fundamental biogeochemical changes that occur
when marine environments become anoxic.

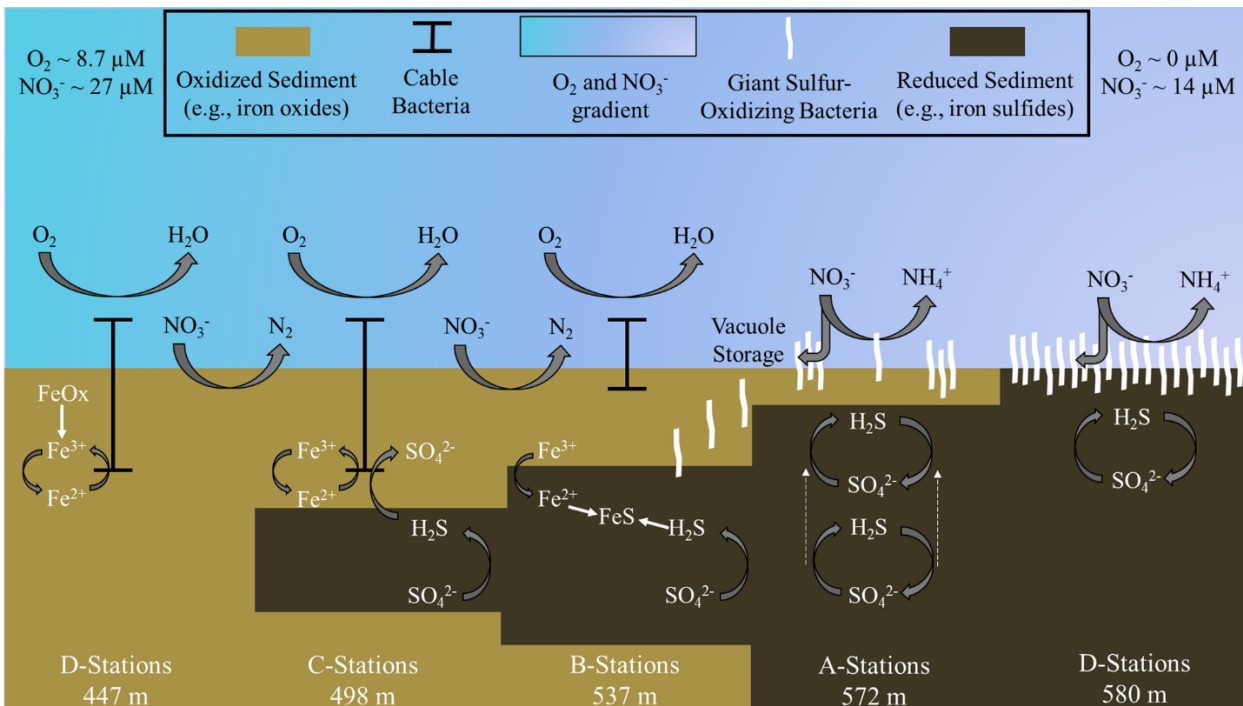

**Figure 7:** Schematic of biogeochemical processes in the Santa Barbara Basin along the depth gradients studied in October/November 2019. Teal to lavender gradient represents a decline in $O_2$ and $NO_3^-$ concentrations with basin depth. In the shallower, hypoxic basin (D-stations), denitrification and iron reduction are dominant and reduced iron is rapidly re-oxidized in near-surface sediment by cable bacteria. Deeper in the basin (A-stations and depocenter), nitrogen cycling shifts towards dissimilatory nitrate reduction to ammonia (DNRA). Reduced iron combines with sulfide, produced by sulfate reduction, diffusing from deeper sediment layers to form iron sulfides. As oxygen concentration approaches zero between the A-stations and the basin's depocenter, giant sulfur-oxidizing bacteria hyper-accumulate nitrate in their intracellular vacuoles. Nitrate removal combined with the exhaustion of available iron oxides in the near-surface sediments allows the zone of sulfate reduction to migrate towards the surface (see dashed arrows at A-stations), providing the giant sulfur-oxidizing bacteria with sufficient reduced sulfur to proliferate into thick, contiguous mats. Note: Figure is not to scale, and processes are simplified to illustrate main concepts.

**Acknowledgements**

We thank the captain, crew, and scientific party of the R/V Atlantis, and the crew of the ROV Jason for their technical and logistical support during the research expedition AT42-19. We also thank Q. Qin, E. Arrington, M. O'Beirne, A. Mazariegos, X. Moreno, A. Eastman, and K. Gosselin for assisting with shipboard analyses. We thank J. Liu for assistance in reviewing the data processing for this manuscript. We further thank M. Alisch from the Max-Planck-Institute in Bremen, Germany for DIC analyses. We thank G. Eickert-Grötzschel, V. Hübner, A. Niclas, I. Schröder, and C. Wigand from the Max-Planck-Institute in Bremen, Germany for constructing the microsensors. We acknowledge J. Matthews from the UC Davis Stable Isotope facility for assisting with solid phase analyses. Funding for this work was provided by the US National Science Foundation, NSF OCE-1756947 and OCE-1830033 (to DLV) and OCE-1829981 (to TT), and a Simons Foundation Postdoctoral Fellowship in Marine Microbial Ecology (No. 547606 to XP). Further support was provided by the Max Planck Society and the Alfred Wegener Institute for Polar and Marine Research.

**Data availability.**

Biogeochemical data presented in this manuscript are accessible through the Biological & Chemical Oceanography Data Management Office (BCO-DMO) at the following landing pages: https://www.bco-dmo.org/dataset/867007; https://www.bco-dmo.org/dataset/867113; https://www.bco-dmo.org/dataset/867221; https://www.bco-dmo.org/dataset/896706

**Author contributions.**
TT, DV, FK, NL, and JT designed the project. DJY, SK, JT, DR, and TT processed sediment
cores at sea. DJY conducted geochemical analyses of sediment porewater and benthic flux
chamber water. DJY prepared TOC and TON samples. DR and SK analyzed sediment porosity
and density. TT and SK performed shipboard sulfate reduction incubations. DJY and DR
conducted sulfate reduction analyses. DJY, NL, and JT transformed and interpreted ROV Jason
data. FJ and FW operated BFC and microprofilers and analyzed associated data. XP conducted
$^{15}$N experiments and analyses. All authors reviewed and edited the manuscript.

**Competing interests.**
At least one of the (co-)authors is a member of the editorial board of Biogeosciences.

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
