# Peer review of "Marine anoxia initiates giant sulfur-bacteria mat proliferation and associated changes in"

_EGUsphere, 2023_

## Referee Comment (RC1)

**Review of "*Marine anoxia initiates giant sulfur-bacteria mat proliferation and associated changes in benthic nitrogen, sulfur, and iron cycling in the Santa Barbara Basin, California Borderland*" by Yousavich et al. BG manuscript egusphere-2023-1198.**

General evaluation

This manuscript reports on a study of benthic bacterial mats in the Santa Barbara Basin, including the prerequisites for their formation, how they develop in response to changing bottom water oxygen conditions (which vary in time and space), and the relationship between presence of mats and benthic Fe, P, N and S cycling.

My major concerns with this submission are:

Previous findings and present knowledge are not taken sufficiently into account. It appears that the authors have not properly checked the literature regarding Fe and N cycling in response to changing bottom water oxygen conditions, and regarding benthic bacterial mats in other settings than the Peru, Mauretanian and Arabian OMZs. The findings of this study can then much better be put in the correct context of previous observations and knowledge, which would considerably improve the manuscript.

There are several papers reporting on the influence of declining bottom water oxygen on benthic Fe and P fluxes. Suggestions for papers to look at and cite are given below. There are of course several other papers on this as well.

The manuscript reports on changes in the relative importance of benthic denitrification and DNRA as bottom water oxygen declines and anoxia is approached. This is fine. I suggest that the authors read present literature on corresponding changes in pathways of benthic NO3-reduction when the bottom water turns the other way, i.e. from anoxia to oxygenated conditions. That would add a very interesting component to the Discussion of this manuscript. Suggestions for papers to look at and cite in this context are given below.

It has not been clarified enough whether this study presents any new findings, or if process patterns and mechanisms in SBB sediments, presented in this study, have previously been found and explored elsewhere. This should be stated clearer in the manuscript.

The Results section is far too long, and too many details are given. Please shorten it and keep only descriptions of the essential parts of the results. The readers also have the figures and SI to get information from.

In several cases the text is not clear enough, and then it has to be clarified. I have given examples of this below, but I encourage the authors to check throughout the manuscript for unclear text / writing.

Specific comments
L 35-37: "*We found that the presence of mats was associated with a shift from denitrification to dissimilatory nitrate reduction to ammonium.*"
Do you mean the development of mats…. (not presence)? A shift is due to a change (such as a development) of something.

Lines (L) 40-42: "*Our research further suggests that cycles of deoxygenation and reoxygenation of the benthic environment result in extremely high benthic fluxes of dissolved iron from the basin's sediment.*"
This or similar findings have been published previously – at several occasions. See e.g. Balzer (1982), Sundby et al. (1986), van de Velde et al. (2020), and papers on the Baltic Sea from the group of Caroline Slomp.
References:
W. Balzer. 1982. On the distribution of iron and manganese at the sediment/water interface: thermodynamic *vs.* kinetic control. Geochim. Cosmochim. Acta 46: 1153-1161.

B. Sundby, L. Anderson, P. Hall, Å. Iverfeldt, M. Rutgers van der Loeff and S. Westerlund. 1986. The effect of oxygen on release and uptake of cobalt, manganese, iron and phosphate at the sediment-water interface. Geochim. Cosmochim. Acta 50: 1281-1288.

S. van de Velde, A. Hylén, M. Kononets, U. Marzocchi, M. Leermakers, K. Choumiline, P. Hall and F. Meysman. 2020. Elevated sedimentary removal of Fe, Mn, and trace elements following a transient oxygenation event in the Eastern Gotland Basin, central Baltic Sea. Geochim. Cosmochim. Acta 271: 16-32. https://doi.org/10.1016/j.gca.2019.11.034

L 60-61: "*and promotes organic matter preservation in the sediments*".
Not necessarily. See e.g. recent paper by van de Velde et al. (2023). Please rephrase this sentence so it better reflects present knowledge.
Reference:
S. van de Velde, A. Hylén, M. Eriksson, R. James, M. Kononets, E. Robertson and P. Hall. 2023. Exceptionally high respiration rates in the reactive surface layer of sediments underlying oxygen-deficient bottom waters. Proc. Royal Society A, in press. https://dx.doi.org/10.1098/rspa.2023.0189.

L 65-85: Would not this paragraph fit better in the Methods section under (e.g.) "Study site"?

L 89: Remove the second "*mats*".

L 115-116: "*suggest that sedimentary organisms are responsible for approximately 75% of the total NO3- uptake in the SBB*"
This sentence needs clarification. For example, what do you mean by uptake? Accumulation in cells? Assimilation by phytoplankton? Do you also include reduction (denitrification, DNRA and anammox) in uptake? What is the remaining 25% of the uptake due to? Please explain.

L 199: "*DIC detection limit was 0.5 mM.*"
Did not Hall and Aller (1992) report a much lower detection limit? If so, why is yours so high?

L 243-244: There is a factor of 1 000 000 between mmol and nmol (not 1 000).

L 282 and 284: "*Where J is the diffusive flux*".
Only diffusive? I would call a flux measured in a benthic chamber a total flux. Even if bottom water has a low oxygen concentration, the sediment may contain bioirrigating animals, so molecular diffusion may not be the only transport mechanism.

L 480-482: "*BFC O2 concentrations were compromised by O2 release from the chamber's polycarbonate walls, which prevented an accurate calculation of O2 fluxes from sensor data.*"
Did not Kononets et al. (2021; which is cited in this manuscript) suggest a way to minimize this problem? Could that procedure not be implemented for your chambers?

L 537-538: "*generally more O2 and NO3- than the southern transect (e.g., 9 μM at NDT3-A and 0 μM at SDT3-A).*"
9 and 0 μM of O2 or of NO3-? Please be comprehensive and clear.

Section 4.1 of Discussion: Benthic microbial mats have also been studied in the central Baltic Sea where they occupy the so called hypoxic transition zone at water depths of about 80-120 m. It would be highly relevant in this manuscript to make comparisons with Baltic benthic mats. The manuscript would benefit from that. See e.g. Noffke et al. (2016), a study which included the influence of these mats on benthic nutrient exchange.
Reference:
A. Noffke, S. Sommer, A. Dale, P. Hall and O. Pfannkuche. 2016. Benthic nutrient fluxes in the Eastern Gotland Basin (Baltic Sea) with particular focus on microbial mat ecosystems. J. Mar. Syst. 158: 1–12. http://dx.doi.org/10.1016/j.jmarsys.2016.01.007

L 585-586: "*…N2 production via denitrification and anammox accounted for 86% of NO3- removal in the shallow basin (NDT3-D, Fig. 6).*"
Is the implication of this that DNRA accounted for 14%? Please clarify.

L 595: "*Declining nitrate concentrations may be as important as anoxia itself to GSOB mat proliferation.*"
Please explain why it may be so. The sentence needs to be clarified. High nitrate concentrations should stimulate GSOB as long as the condition is oxygen-free. Then, the sentence does not make sense in my mind.

L 611-612: "*While low-nitrate conditions could benefit GSOB mats,…*".
Why? How? This has not been clarified. Please explain.

Section 4.2: There are published studies on the effect of oxygenation of bottom water of anoxic basins on benthic nitrogen cycling including nitrate reduction (especially denitrification and DNRA). That is, going the opposite way (from anoxia to oxygenated conditions) than what this manuscript studies (from oxygenated conditions to anoxia). I think the Discussion of the manuscript would much improve by including comparisons with such studies, e.g. De Brabandere et al. (2015) and Hylén et al. (2022). Topics to discuss may include: Are patterns reversed when one goes from condition A to B compared to from condition B to A? Are systems reversible? See also my comments on this above.
References:
L. De Brabandere, S. Bonaglia, M. Kononets, L. Viktorsson, A. Stigebrandt, B. Thamdrup and P. Hall. 2015. Oxygenation of an anoxic fjord basin strongly stimulates benthic denitrification and DNRA. Biogeochemistry 126: 131–152. DOI 10.1007/s10533-015-0148-6

A. Hylén, S. Bonaglia, E. Robertson, U. Marzocchi, M. Kononets and P. Hall. 2022. Enhanced benthic nitrous oxide and ammonium production after natural oxygenation of long-term anoxic sediments. Limnol. Oceanogr. 67, 419–433. doi: 10.1002/lno.12001

L 658: "*mechanics*".
Is mechanisms not a better term?

Section 4.3: Please clearer specify if this study presents any new findings, or if SBB sediments display processes and mechanisms (relevant for this section) which previously have been found and explored elsewhere.

L 661: Section 4.3 appears here a second time. Please rename to 4.4

L 693-694: "*While sulfate reduction rates for B-stations are absent,*"
Do you mean that SRR were not measured, or were they measured and found to be zero? Please clarify.

Second Section 4.3 (which should be called 4.4): There is a lot of text reporting raw results. Can this text be shortened and moved to Results? It fits better there than in Discussion.

L 744: Section 4.4 should be renamed to 4.5.

L 759-760: "*These analogous observations highlight the importance of alternating redox conditions to establish high benthic iron fluxes.*"
This observation has been found and published several times previously. Please better check the literature and put your observations in a correct context of present knowledge. That will improve this manuscript. See also my comments in the beginning of this review.

L 784-785: "*We found that GSOB mats proliferate in the SBB where the bottom water is anoxic and nitrate concentrations are declining, ...*"
It has not been explained or clarified why GSOB mats proliferate when $NO_3^-$ concentrations go down. Please explain.

L 798-791: "*We conclude that changes in iron minerology, specifically the formation of an iron sulfide layer deeper in sediments, encourages the elevation of the sulfate reduction zone.*"
Which is the underlying reason for this, and which is the mechanism? This has not been clarified enough. Please explain and clarify.

L 795-796: "*Further, the same transient deoxygenation that allows for these mats to flourish also allows for a high $Fe^{2+}$ and $PO_4^{3-}$ flux into the SBB water column.*"
I have commented on the finding of elevated Fe fluxes above.
Already Mortimer (1941) showed that elevated phosphate fluxes from sediment occur upon deoxygenation of the bottom water. Again, please put your findings in the adequate context of present knowledge and previous findings. See also my comments on this above.

L 942-945: This reference is incompletely typed.

I recommend that this submission should undergo a major revision, taking my comments above into account, before it can be considered for acceptance.

---

## Author Response (AR1)

**Dear Dr. Slomp,**

Please see our revised manuscript that incorporated the requested changes from editors and went through further editing from all co-authors. We would like to highlight two changes. We included a statement in section 4.2 about the contribution of meiofauna and diatoms to benthic nitrate reduction. Additionally, we found an error in the way the TOC results were calculated that changed our averaged numbers detailed in the results, discussion, and table 2. These changes didn't substantially change our interpretations of the results, however.

**REVIEWER 1**

We like to thank Referee 1 for their helpful feedback on our manuscript. In the following, we provide a point-by-point response to the referee's comments.

Referee 1: Review of "Marine anoxia initiates giant sulfur-bacteria mat proliferation and associated changes in benthic nitrogen, sulfur, and iron cycling in the Santa Barbara Basin, California Borderland" by Yousavich et al.  BG manuscript egusphere-2023-1198.

General evaluation

This manuscript reports on a study of benthic bacterial mats in the Santa Barbara Basin, including the prerequisites for their formation, how they develop in response to changing bottom water oxygen conditions (which vary in time and space), and the relationship between presence of mats and benthic Fe, P, N and S cycling.

My major concerns with this submission are:

Previous findings and present knowledge are not taken sufficiently into account. It appears that the authors have not properly checked the literature regarding Fe and N cycling in response to changing bottom water oxygen conditions, and regarding benthic bacterial mats in other settings than the Peruvian, Mauritanian, and Arabian OMZs. The findings of this study can then much better be put in the correct context of previous observations and knowledge, which would considerably improve the manuscript.

Authors' Response:  We placed our study within the context of other marine low oxygen environments (Peruvian OMZ, Mauritanian OMZ, Namibian OMZ, Arabian OMZ). We do acknowledge, however, that several studies presented in the referee's comments, particularly those in the Baltic Sea, have relevant importance to our study and will incorporate them into the discussion. For example, text detailing the appearance of mats in the "hypoxic transition zone" of the Eastern Gotland Basin, rather than the anoxic parts of the basin, will be added to section 4.1. A tentative draft of the text reads:

"Interestingly, GSOB mats in the Eastern Gotland Basin of the Baltic Sea were confined to a hypoxic transition zone, where O2 was < 30 µM but did not reach anoxia, while no mats were observed at deeper anoxic locations (Noffke et al., 2016). This difference in distribution compared to the SBB suggests that GSOB mats proliferate under different conditions (anoxic or hypoxic), potentially depending on the species of mat-forming bacteria present and whether they specialize in aerobic or anaerobic chemosynthesis."

Referee 1: There are several papers reporting on the influence of declining bottom water oxygen on benthic Fe and P fluxes. Suggestions for papers to look at and cite are given below. There are of course several other papers on this as well.

Authors' Response:  We are thankful for the additional references and will incorporate some of them where appropriate to complement and strengthen the results reported here and put additional context to their discussion.

Referee 1: The manuscript reports on changes in the relative importance of benthic denitrification and DNRA as bottom water oxygen declines and anoxia is approached. This is fine. I suggest that the authors read present literature on corresponding changes in pathways of benthic NO3- reduction when the bottom water turns the other way, i.e. from anoxia to oxygenated conditions. That would add a very interesting component to the Discussion of this manuscript. Suggestions for papers to look at and cite in this context are given below.

Authors' Response:  We like to thank the referee for their suggestions. We will incorporate the suggested literature and include a discussion about the differences between shifts to and from anoxia with respect to nitrogen cycling. Comparing our data to De Brabandere et al., 2015 and Hylén et al., 2022 shows that DNRA thrives in transiently deoxygenated systems and is resilient to weak reoxygenation events, only yielding to denitrification during prolonged reoxygenation events that increase oxygen concentrations above hypoxia. A tentative draft of our text reads:

"While our study suggests a shift from denitrification to DNRA during deoxygenation of SBB bottom water, other studies examined changes in benthic nitrogen cycling under reverse conditions, i.e., the reoxidation of the environment following anoxia  (Hylén et al., 2022; De Brabandere et al., 2015). After a decadal oxygenation event in the Eastern Gotland Basin (Baltic Sea) in 2015-2016, sediment exhibited a slight increase in denitrification, but remained dominated by DNRA and N2O production (Hylén et al., 2022). The lack of N2 production via denitrification following this oxygenation event was attributed to the reoxygenation event being too weak to substantially oxidize sediments, which would favor denitrification (Hylén et al., 2022). In an engineered reoxygenation event of the By Fjord on Sweden's western coast, where dissolved O2 and NO3- content of anoxic and antiric bottom water was artificially increased to approx. 130 µM O2 and 20 µM NO3- over a period of roughly 2 years, denitrification rates were increased by an order of magnitude and DNRA rates were also stimulated (De Brabandere et al., 2015). Comparing our results to these two studies suggests that organisms that do DNRA are more resilient to weak reoxygenation events and thrive in transiently deoxygenated systems that remain hypoxic (O2 < 63 µM). The frequency and magnitude of reoxygenation and deoxygenation of SBB bottom waters, and the effect of these processes on the benthic microbial community, could be a major factor supporting some of the highest recorded total nitrate reduction rates in a natural benthic marine setting (Peng et al., 2023)."

Referee 1: It has not been clarified enough whether this study presents any new findings, or if process patterns and mechanisms in SBB sediments, presented in this study, have previously been found and explored elsewhere. This should be stated clearer in the manuscript.

Authors' response:  Thank you for bringing this omission to our attention. We will add a few sentences at the end of the introduction to highlight the novelty of our study. A tentative draft of the text reads:

"These investigations represent the first basin-wide geochemical characterization of the Santa Barbara Basin which hosts the largest as-of-yet mapped GSOB mat in the world's oceans. It is the first suite of in-situ flux measurements carried out in the SBB, which is unique to other heavily studied marine settings (e.g., Eastern Gotland Basin, Peruvian upwelling zone) in that it is an oceanic basin within an upwelling zone. The results presented here also provide geochemical context for a number of other related investigations in the SBB (Robinson et al., 2022; Peng et al., 2023) as well as the first measurements in a multi-year study of biogeochemical changes in response to warming waters and increased stratification on the California coast."

Referee 1: The Results section is far too long, and too many details are given. Please shorten it and keep only descriptions of the essential parts of the results. The readers also have the figures and SI to get information from.

Authors' Response:  Thank you for this advice. We will trim down the results section of the revised version by reducing redundancy (e.g., duplicative sediment lamination descriptions), removing unnecessary detail (e.g., referring to fewer stations when giving examples), and removing unnecessary descriptions of data (e.g., changes between core supernatant and porewater solute concentrations).

Referee 1: In several cases the text is not clear enough, and then it has to be clarified. I have given examples of this below, but I encourage the authors to check throughout the manuscript for unclear text / writing.

Authors' response:  We thank the reviewer for pointing us to these examples that need clarifications. We will revise the text accordingly as suggested below. We are unsure what the reviewer means when they state "in several cases the text is not clear enough, and then it has to be clarified." Based on the examples provided, we interpret this to mean:

1) Several long explanations could be reduced by choosing more appropriate words.
2) There are places where more appropriate citations could be included.
3) There are places where grammatical fixes are needed.
4) Some processes (e.g., iron sulfide formation, DNRA vs denitrification at different nitrate concentrations) are not thoroughly explained.

Assuming these are the areas of concern, we will go over the entire manuscript to improve clarity.

Specific comments

Referee 1: L 35-37: "We found that the presence of mats was associated with a shift from denitrification to dissimilatory nitrate reduction to ammonium."

Do you mean the development of mats…. (not presence)? A shift is due to a change (such as a development) of something.

Authors' Response:  The referee is correct that 'development of mats' would be more precise. We will change the wording to "development". Thanks!

Referee 1: Lines (L) 40-42: "Our research further suggests that cycles of deoxygenation and reoxygenation of the benthic environment result in extremely high benthic fluxes of dissolved iron from the basin's sediment."

This or similar findings have been published previously – at several occasions. See e.g. Balzer (1982), Sundby et al. (1986), van de Velde et al. (2020), and papers on the Baltic Sea from the group of Caroline Slomp.

References:

1. Balzer. 1982. On the distribution of iron and manganese at the sediment/water interface: thermodynamic vs. kinetic control. Geochim. Cosmochim. Acta 46: 1153-1161.

1. Sundby, L. Anderson, P. Hall, Å. Iverfeldt, M. Rutgers van der Loeff and S. Westerlund. 1986. The effect of oxygen on release and uptake of cobalt, manganese, iron and phosphate at the sediment-water interface. Geochim. Cosmochim. Acta 50: 1281-1288.

1. van de Velde, A. Hylén, M. Kononets, U. Marzocchi, M. Leermakers, K. Choumiline, P. Hall and F. Meysman. 2020. Elevated sedimentary removal of Fe, Mn, and trace elements following a transient oxygenation event in the Eastern Gotland Basin, central Baltic Sea. Geochim. Cosmochim. Acta 271: 16-32. https://doi.org/10.1016/j.gca.2019.11.034

Authors' Reply: Thanks a lot for these suggestions. Note, however, that Balzer et al. 1982 utilized a bell jar to manipulate benthic conditions into anoxia over a 99-day experiment. While the findings in this paper are foundational to benthic flux estimates, the experiment conducted, the benthic setting (bell jar depth of 20 m), and the conditions (99-day experiment with multiple samplings) described are different to our study. Similarly, Sundby et al. 1986 is a foundational paper about benthic element fluxes, but the experimental design involved manipulating the environment to study these effects, and the study-site was within the photic zone (6 m depth). Thus, we do not find the results of Sundby et al. 1986 necessary for citation either. However, we will include a citation to Mortimer, 1941 at the beginning of section 4.5, which predates both studies, and which we find to be a more appropriate source for benthic iron and phosphate fluxes into anoxic waters.

We agree that van de Velde et al. 2020 is a great analogous study to ours in a (somewhat) similar coastal basin environment. In that paper, the flux caused by the reduction of oxidized minerals (termed "enhanced elemental recycling") at the sediment-water interface is attributed to oxide mineral formation in the water column. These minerals then sink back to the anoxic sediment, where they are subsequently reduced, providing a flux back into the water column. While that certainly could be a component in the Santa Barbara Basin as well, the theory postulated here (built off previous speculation in Kuwabara et al. 1999 and backed by our observations during the 2019 expedition) is that seasonal re-oxygenation of bottom waters directly oxidizes the sediment-water interface. The end-result is the same (an oxidized surface layer that is subsequently reduced, providing a flux into anoxic bottom water) but the mechanism proposed is slightly different. The differences in mixing and basin depth between the

Santa Barbara Basin and the Eastern Gotland Basin are important here, as the former experiences frequent (<1 year) cycles of reoxygenation and deoxygenation, whereas the latter experiences reoxygenation events on the scale of decades. Interestingly, the fluxes recorded in van de Velde et al. are also approx. 2 orders of magnitude smaller than the fluxes recorded in the Santa Barbara Basin. We believe this provides further evidence that differences within the frequency of reoxygenation and physical setting (both geomorphology and mixing) effect the magnitude of benthic iron and phosphate fluxes. We will expand our discussion with a reference to the van de Velde 2020 study.

Kuwabara, J. S., van Geen, A., McCorkle, D. C., and Bernhard, J. M.: Dissolved sulfide distributions in the water column and sediment pore waters of the Santa Barbara Basin, Geochimica et Cosmochimica Acta, 63, 2199-2209, 1999.

Referee 1: L 60-61: "and promotes organic matter preservation in the sediments".

Not necessarily. See e.g. recent paper by van de Velde et al. (2023). Please rephrase this sentence so it better reflects present knowledge.

Reference:

1. van de Velde, A. Hylén, M. Eriksson, R. James, M. Kononets, E. Robertson and P. Hall. 2023. Exceptionally high respiration rates in the reactive surface layer of sediments underlying oxygen-deficient bottom waters. Proc. Royal Society A, in press. https://dx.doi.org/10.1098/rspa.2023.0189.

Authors' response:  Thank you for pointing us to this very recent study, whose publication was subsequent to the publication of our preprint. We will add the new hypothesis to the introduction to capture the current scientific discussion. We, however, like to point out that the finding of van de Velde et al. 2023 has no impact on the interpretation of our data. van de Velde et al. 2023 postulate that a thin reactive surface layer, featuring high in-situ DIC fluxes, is present in laminated sediment below anoxic water. Different to their observation, we found no detectable evidence of DIC or total alkalinity fluxes from the sediment into the water column based on our in-situ chamber incubations (data not shown). Probably, longer in-situ incubations might have been sufficiently sensitive to detect changes in DIC or TA in the chamber water; however, these fluxes would still be very low. We therefore believe that the Santa Barbara Basin does not feature a reactive surface layer as described in van Velde et al. Given this new scientific discussion, we decided to add our Total Alkalinity and DIC data to the results/supplementary.

Referee 1: L 65-85: Would not this paragraph fit better in the Methods section under (e.g.) "Study site"?

Reply:  Thank you for your suggestion. We will move geographic/hydrographic/biogeochemical details about the Santa Barbara Basin into a new "study site" section within the Methods and trim the introduction to the most relevant information.

Referee 1: L 89: Remove the second "mats".

Authors' Response:  We will remove accordingly. Thanks!

Referee 1: L 115-116: "suggest that sedimentary organisms are responsible for approximately 75% of the total NO3- uptake in the SBB"

This sentence needs clarification. For example, what do you mean by uptake? Accumulation in cells? Assimilation by phytoplankton? Do you also include reduction (denitrification, DNRA and anammox) in uptake? What is the remaining 25% of the uptake due to? Please explain.

Authors' response: This sentence references Sigman et al. 2003, in which water column nitrate depletion in the Santa Barbara Basin water column was studied using nitrogen and oxygen isotope analyses and compared to similar nitrate depletion in an open ocean OMZ. Sigman et al. 2003 found that 75% of "nitrate loss" in the Santa Barbara Basin was due to benthic denitrification. However, the authors note in Section 5 ("Implications") that they cannot distinguish between canonical denitrification and other nitrate reduction pathways. We noticed that "nitrate uptake" may be too vague and will therefore change the wording to "nitrate reduction" to clarify. The other approx. 25% of nitrate reduction occurs in the water column, which we feel needs no further explanation as the reader can logically come to that conclusion.

Sigman, D. M., Robinson, R., Knapp, A., Van Geen, A., McCorkle, D., Brandes, J., and Thunell, R.: Distinguishing between water column and sedimentary denitrification in the Santa Barbara Basin using the stable isotopes of nitrate, Geochemistry, Geophysics, Geosystems, 4, 2003.

Referee 1: L 199: "DIC detection limit was 0.5 mM."

Did not Hall and Aller (1992) report a much lower detection limit? If so, why is yours so high?

Author's Response:  We reported a detection limit of 0.5 mmol/L as this was the concentration of the lowest standard that was used to build the calibration curve. However, we will report the methodological limit of 0.1 mmol/L as reported in Hall and Aller (1992) to avoid confusion. Thanks!

L 243-244: There is a factor of 1 000 000 between mmol and nmol (not 1 000).

Authors' response:  Thanks for noticing. The conversion is actually from mmol $L^{-1}$ to nmol $cm^{-3}$ so the conversion factor is correct. However, we decided to delete this methodological detail to shorten the manuscript as we precisely followed the protocol of Kallmeyer et al. 2004.

Referee 1: L 282 and 284: "Where J is the diffusive flux".

Only diffusive? I would call a flux measured in a benthic chamber a total flux. Even if bottom water has a low oxygen concentration, the sediment may contain bioirrigating animals, so molecular diffusion may not be the only transport mechanism.

Authors' Response:  Thanks for noticing. We will delete the word "diffusive".

Referee 1: L 480-482: "BFC O2 concentrations were compromised by O2 release from the chamber's polycarbonate walls, which prevented an accurate calculation of O2 fluxes from sensor data."

Did not Kononets et al. (2021; which is cited in this manuscript) suggest a way to minimize this problem? Could that procedure not be implemented for your chambers?

Authors' response:   The recommendation provided by Kononets et al. (2021) in section 2.5 ('Control of incubation functioning using oxygen sensors') is (1) to monitor possible oxygen concentration from plastics in the chambers with oxygen optodes and (2) to keep the chamber lid open for the time required to avoid oxygenation of the incubated water (typically 4h according to their experience).

We followed the first recommendation and that made us aware of the oxygen release from the polycarbonate walls that we report in our manuscript. We could not follow their second recommendation, though. The chambers we used do not have an open lid that can be closed after deployment (the lid has openings to release excess water when chambers are being pushed into the sediment). In the interest of deployment time and replication of the chamber incubations we also decided against alternative options (e.g., leaving the chambers on the elevator exposed to bottom water before deployment at the seafloor).

Referee 1: L 537-538: "generally more O2 and NO3- than the southern transect (e.g., 9 µM at NDT3-A and 0 µM at SDT3-A)."

9 and 0 µM of O2 or of NO3-? Please be comprehensive and clear.

Authors' response:  We will clarify that the quoted numbers are in reference to oxygen.

Referee 1: Section 4.1 of Discussion: Benthic microbial mats have also been studied in the central Baltic Sea where they occupy the so-called hypoxic transition zone at water depths of about 80-120 m. It would be highly relevant in this manuscript to make comparisons with Baltic benthic mats. The manuscript would benefit from that. See e.g., Noffke et al. (2016), a study which included the influence of these mats on benthic nutrient exchange.

Reference:

1. Noffke, S. Sommer, A. Dale, P. Hall and O. Pfannkuche. 2016. Benthic nutrient fluxes in the Eastern Gotland Basin (Baltic Sea) with particular focus on microbial mat ecosystems. J. Mar. Syst. 158: 1–12. http://dx.doi.org/10.1016/j.jmarsys.2016.01.007

Authors' response:  Thank you for pointing us to this publication, it provides several additional comparisons that proved illuminating. We will reference this study in the introduction. We will further include a discussion of the mat distribution relative to bottom water oxygen content in section 4.1, a discussion of DNRA and the magnitude of ammonium flux in section 4.2, and a discussion of the magnitude of phosphate flux in section 4.5.

Referee 1: L 585-586:  "…N2 production via denitrification and anammox accounted for 86% of NO3- removal in the shallow basin (NDT3-D, Fig. 6)."

Is the implication of this that DNRA accounted for 14%? Please clarify.

Authors' response:  Thanks for pointing us to the missing information. We will change the wording to clarify that removal in this context only refers to "nitrate reduction" - and then provide the respective percentages of the processes involved. We will further add a sentence to clarify that we are not including $NO_3^-$ uptake (via vacuoles) in this calculation.

Referee 1: L 595: "Declining nitrate concentrations may be as important as anoxia itself to GSOB mat proliferation."

Please explain why it may be so. The sentence needs to be clarified. High nitrate concentrations should stimulate GSOB as long as the condition is oxygen-free. Then, the sentence does not make sense in my mind.

Authors' Response: Please see our response to the next question.

Referee 1: L 611-612: "While low-nitrate conditions could benefit GSOB mats,…".

Why? How? This has not been clarified. Please explain.

Authors' Response: Thank you for the suggestion. We will include more precise citations and in-text references to studies that have examined the thermodynamics between DNRA and denitrification at different electron donor to acceptor ratios. We will further add a table (Table 3) with the Gibbs free energies for metabolisms relevant (DNRA with nitrate/sulfide and denitrification with nitrate/TOC) for this discussion. We are further working on integrating a calculation of the Gibbs free energy of denitrification coupled to acetate oxidation based on acetate concentrations determined in a study recently accepted in Biogeosciences, which was done in tandem with our work (Krause et al., in press).

A tentative draft of the text reads:

"A high ratio of electron donor to electron acceptor favors DNRA over denitrification  (Marchant et al., 2014; Hardison et al., 2015; Tiedje et al., 1983) and this ratio appears to be critical in determining the dominant nitrate reduction pathway in SBB sediments, similar to the Eastern Gotland Basin (Hylén et al., 2022) and the By Fjord (De Brabandere et al., 2015). The energy yields for denitrification and DNRA at environmental conditions in the SBB depocenter (6°C, 58 atm) are shown in Table 3.  As discussed in (Tiedje et al., 1983), heterotrophic denitrification yields more energy per mol of electron donor than DNRA. However, the reverse is true when considering energy yield per mol of electron acceptor (NO3-). DNRA also yields 3 more electrons per molecule of NO3- than denitrification. Tiedje et al. argued that in environments that are starved of powerful terminal electron acceptors, (e.g., oxygen, nitrate) such as anoxic, organic-rich sediment, the energy yield per electron acceptor and additional electrons available for transfer, could push nitrate reduction towards DNRA. Multiple laboratory studies have converged on an electron donor to acceptor ratio of approx. 3 to encourage DNRA over denitrification (Hardison et al., 2015; Algar and Vallino, 2014) though other studies have found higher values (Porubsky et al., 2009; Kraft et al., 2014). Our TOC results (Table 2) suggest that bottom water nitrate concentrations would need to drop to approx. 2 µM before DNRA would be more favorable, however, the sulfide concentrations near the sediment-water interface at the SBB depocenter (approx. 200 µM at 0.5 cm depth; Fig. 3, NDRO) would favor chemoautotrophic DNRA over denitrification at ambient marine nitrate concentrations (approx. 28 µM). Additionally, DNRA appears to be the preferred nitrate reduction pathway for chemoautotrophs that utilize iron or sulfide as an electron donor (Caffrey et al., 2019;

Kessler et al., 2019; An and Gardner, 2002). As GSOB mats hyper-accumulate nitrate from the bottom water into their intracellular vacuoles, the resulting decline in electron acceptors at the sediment-water interface coupled with an elevation of the sulfate reduction zone would create an electron donor to acceptor ratio that favors DNRA. Since GSOB mats in the SBB seem to prefer DNRA, starving the bottom water of electron acceptors coupled with the high sulfate reduction rates afforded by the sediment TOC content could give them a competitive advantage and allow them to proliferate into the largest-yet mapped GSOB mat in Earth's oceans, as seen in other expeditions (Valentine et al., 2016; Reimers et al., 1996b; Kuwabara et al., 1999)."

Tiedje, J. M., Sexstone, A. J., Myrold, D. D., and Robinson, J. A.: Denitrification: ecological niches, competition and survival, Antonie van Leeuwenhoek, 48, 569-583, 1983.

Krause, S. J. E., Liu, J., Yousavich, D. J., Robinson, D., Hoyt, D., Qin, Q., Wenzhoefer, F., Janssen, F., Valentine, D., and Treude, T.: Evidence of cryptic methane cycling and non-methanogenic methylamine consumption in the sulfate-reducing zone of sediment in the Santa Barbara Basin, California, EGUsphere [preprint], https://doi.org/10.5194/egusphere-2023-909, 2023.

Referee 1: Section 4.2: There are published studies on the effect of oxygenation of bottom water of anoxic basins on benthic nitrogen cycling including nitrate reduction (especially denitrification and DNRA). That is, going the opposite way (from anoxia to oxygenated conditions) than what this manuscript studies (from oxygenated conditions to anoxia). I think the Discussion of the manuscript would much improve by including comparisons with such studies, e.g. De Brabandere et al. (2015) and Hylén et al. (2022). Topics to discuss may include: Are patterns reversed when one goes from condition A to B compared to from condition B to A? Are systems reversible? See also my comments on this above.

References:

1. De Brabandere, S. Bonaglia, M. Kononets, L. Viktorsson, A. Stigebrandt, B. Thamdrup and P. Hall. 2015. Oxygenation of an anoxic fjord basin strongly stimulates benthic denitrification and DNRA. Biogeochemistry 126: 131–152. DOI 10.1007/s10533-015-0148-6

1. Hylén, S. Bonaglia, E. Robertson, U. Marzocchi, M. Kononetsand P. Hall. 2022. Enhanced benthic nitrous oxide and ammonium production after natural oxygenation of long-term anoxic sediments. Limnol. Oceanogr. 67, 419–433. doi: 10.1002/lno.12001

Reply: We like to thank the referee for their suggestions. As also mentioned above, we will incorporate the suggested literature and include a discussion about the differences between shifts to and from anoxia with respect to nitrogen cycling. Comparing our data to (De Brabandere et al., 2015) and (Hylén et al., 2022) shows that DNRA thrives in transiently deoxygenated systems and is resilient to weak reoxygenation events, only yielding to denitrification during prolonged reoxygenation events that increase oxygen concentrations above hypoxia. A draft of the tentative text we plan to add is found above.

Referee 1: L 658: "mechanics".

Is mechanisms not a better term?

Authors' response:  Agreed, we will change to "mechanisms". Thanks!

Referee 1: Section 4.3: Please clearer specify if this study presents any new findings, or if SBB sediments display processes and mechanisms (relevant for this section) which previously have been found and explored elsewhere.

Author's Response: Thank you for bringing this omission to our attention. As mentioned above we will add a few sentences at the end of the introduction to highlight the novelty of our study. A tentative draft of the text we plan to add is found above.

Referee 1: L 661: Section 4.3 appears here a second time. Please rename to 4.4

 Authors' Response:  We will change accordingly, thanks!

Referee 1: L 693-694: "While sulfate reduction rates for B-stations are absent,"

Do you mean that SRR were not measured, or were they measured and found to be zero? Please clarify.

 Authors' Response: Sulfate reduction samples at B-stations were subject to technical issues in the laboratory and therefore the data is absent. We will add language in the appropriate section to add more clarity.

Referee 1: Second Section 4.3 (which should be called 4.4): There is a lot of text reporting raw results. Can this text be shortened and moved to Results? It fits better there than in Discussion.

Authors' Response:  We assume this comment refers to paragraph 2 of section 4.4. We will move portions of section 4.4, paragraph 2 from the discussion to the results and rewrite portions of this paragraph to focus on interpretations. Thanks for the suggestion!

Referee 1: L 744: Section 4.4 should be renamed to 4.5.

Authors' Response:  Will do accordingly, thanks!

Referee 1: L 759-760: "These analogous observations highlight the importance of alternating redox conditions to establish high benthic iron fluxes."

This observation has been found and published several times previously. Please better check the literature and put your observations in a correct context of present knowledge. That will improve this manuscript. See also my comments in the beginning of this review.

Authors' Response:  Thank you for your input. We will rephrase this concluding sentence to clarify that we are not claiming a novel finding, but rather that our study further emphasizes and complements the findings of other studies referred to in this paragraph. Additionally, we will add text in this paragraph

referring to the work done in the Eastern Gotland Basin as described in van de Velde et al. 2020 to provide a comparison to the Baltic Sea that compliments the existing comparison to the Peruvian margin. A tentative draft of the text reads:

"In a different study from the Eastern Gotland Basin in the Baltic Sea, enhanced iron fluxes were observed during a decadal oxygen flushing event (Van De Velde et al., 2020), which was attributed to enhanced elemental recycling, or cycles of mineral precipitation in the water column followed by mineral dissolution once those minerals sunk to the anoxic sediment. These analogous observations between our study and those detailed above further emphasizes the importance of alternating redox conditions to establish high benthic iron fluxes."

Referee 1: L 784-785: "We found that GSOB mats proliferate in the SBB where the bottom water is anoxic and nitrate concentrations are declining, …"

It has not been explained or clarified why GSOB mats proliferate when $NO_3^-$ concentrations go down. Please explain.

Authors' Response: We will include more precise citations and in-text references to studies that have examined the thermodynamics between DNRA and denitrification at different electron donor to acceptor ratios (please see a tentative text draft above). We hope this clarifies the matter enough to be included in our conclusion.

Referee 1: L 798-791: "We conclude that changes in iron minerology, specifically the formation of an iron sulfide layer deeper in sediments, encourages the elevation of the sulfate reduction zone."

Which is the underlying reason for this, and which is the mechanism? This has not been clarified enough. Please explain and clarify.

Authors' Response: The underlying mechanism, as detailed in paragraph 4 of section 4.4, is that cable bacteria build up an iron oxide layer near the sediment water interface through iron and/or sulfide oxidation (Seitaj et al., 2015). Upon deoxygenation, cable bacteria are replaced by GSOB mats, and the iron oxide buffer is exhausted through iron reduction. Both our porewater geochemistry and visual analyses of sediment conclude that this process is occurring as we transition from the shallower stations under hypoxic bottom water, to deeper stations under anoxic bottom water.

The way this conclusion was written, however, was a bit misleading. We will reword it to clarify that the exhaustion of the iron oxide layer in upper sediments allows for the sulfate reduction zone to migrate upwards, and that this process is critical for mat proliferation. We do not have DNA results for this manuscript, so we cannot definitively say whether the exhaustion of iron oxides we see as we transition deeper into the basin is due to the disappearance of cable bacteria or not, but that is the hypothesis we present in the discussion.

Referee 1: L 795-796: "Further, the same transient deoxygenation that allows for these mats to flourish also allows for a high $Fe^{2+}$ and $PO_4^{3-}$ flux into the SBB water column."

I have commented on the finding of elevated Fe fluxes above.

Already Mortimer (1941) showed that elevated phosphate fluxes from sediment occur upon deoxygenation of the bottom water. Again, please put your findings in the adequate context of present knowledge and previous findings. See also my comments on this above.

Authors' Response:  Please note that you are referring to the final conclusion of the manuscript, which wraps up our findings and interpretations regarding the Santa Barbara Basin (not OMZs in general). We do not claim to be the first to observe elevated iron and phosphate fluxes from OMZ sediment using benthic chamber incubations.  But we report, for the first time, such fluxes from the Santa Barbara Basin. And we provide hypotheses on why the observed iron and phosphate fluxes are higher compared to similar settings.

We appreciate the suggestion to read Mortimer 1941, which we will add to this paper at the beginning of Section 4.5.

Referee 1: L 942-945: This reference is incompletely typed.

Authors' response:  We are unsure which omission the referee is referring to. All references were downloaded and prepared using Endnote's "biogeosciences" style. We could not identify incomplete information in the citation.

Referee 1: I recommend that this submission should undergo a major revision, taking my comments above into account, before it can be considered for acceptance.

Authors' Response:  We like to thank the Referee again for their time invested to provide comments on our manuscript and hope that we addressed shortcomings sufficiently.

**REVIEWER 2**

We like to thank Referee 2 for their helpful feedback on our manuscript. In the following, we provide a point-by-point response to the referee's comments.

Referee 2: The authors investigate how environmental conditions control the proliferation of giant sulphur-oxidizing bacteria by comparing stations with contrasting bottom water redox conditions. Their findings are mostly based on pore water profiles complimented with radiotracer experiments and bulk solid-phase data. Unfortunately, there are no data available on genomics and bacterial abundances.

Authors' Response: Genomic and molecular biology studies that complement this work will be published in a separate study. Given the complexity of datasets already presented in the current study, molecular work would significantly extend the length and dilute the focus of this study. Regardless, this work represents the first basin-wide geochemical study of the Santa Barbara Basin, and the first benthic flux chamber experiments conducted in the Santa Barbara Basin.

Referee 2: My major concern with this manuscript is that the findings need to be put in more context and the underlying mechanisms in certain processes need to be explained in greater detail.

Authors' Response: We acknowledge that missing context was a major shortcoming of the originally submitted manuscript (also noted by Referee 1). We hope that we addressed the Referee's concerns, and we plan to provide more context in our revisions.

Referee 2: 1) What findings are truly novel?

Authors' Response: We will add a few sentences at the end of the introduction to highlight the novelty of this study. A tentative draft of the text reads:

"These investigations represent the first basin-wide geochemical characterization of the Santa Barbara Basin which hosts the largest as-of-yet mapped GSOB mat in the world's oceans. It is the first suite of in-situ flux measurements carried out in the SBB, which is unique to other heavily studied marine settings (e.g., Eastern Gotland Basin, Peruvian upwelling zone) in that it is an oceanic basin within an upwelling zone. The results presented here also provide geochemical context for a number of other related investigations in the SBB (Robinson et al., 2022; Peng et al., 2023) as well as the first measurements in a multi-year study of biogeochemical changes in response to warming waters and increased stratification on the California coast."

Referee 2: 2) Which findings are unique to the Santa Barbara Basin and how do they compare to other systems?

Authors' Response: We detail in the discussion that the magnitude of fluxes seen in the SBB is far higher than those measured in similar transiently deoxygenated environments like the Peruvian OMZ. Additionally, we plan to add comparisons to the Eastern Gotland Basin and other deoxygenated sites within the Baltic Sea that reinforce the similarities and difference to similar environments in sections 4.2 and 4.5. For more details please see our response to Referee 1, who suggested to integrate a couple of studies to our discussions.

Referee 2: 3) To what extent are the observed biogeochemical transformations attributable to the giant sulphur-oxidizing bacteria?

Authors' Response:

Since we do not report DNA or molecular biology experiments in this manuscript, we are cautious with how much we want to attribute specifically to the giant sulfur-oxidizing bacteria (GSOB). Some of the main observed biogeochemical transformations attributable to the GSOB mats are:

1) Total nitrate uptake by sediments is unaffected by mat presence, but they are associated with increased ammonium flux where they appear deeper in the basin. We will revise text in Section 4.2 to reflect this. A tentative draft of this text reads:

   "While total benthic nitrate uptake remained unchanged based on in-situ $NO_3^-$ flux measurements, $NH_4^+$ release from the sediment into the water column increased where GSOB mats were present (Fig. 5).

2) GSOB mats may encourage the sulfate reduction zone to rise via their hyper-accumulation of nitrate, but mat presence alone does not explain this phenomenon, as evidenced by our results in the NDT3-C station. We will revise the text in the conclusion. A tentative draft of this text reads:

   "The hyper-accumulation of $NO_3^-$ by GSOB mats potentially facilitates sulfate reduction close to the sediment-water interface in the SBB (e.g., NDRO and NDT3-A as seen in fig. 2N and 2O) by starving the sediment of this more powerful electron acceptor. The rise of the sulfate reduction zone at NDT3-C (fig. 2L) further suggests that the exhaustion of iron oxides and the formation of iron sulfide below the sediment-water interface may play a crucial role in controlling the distribution of sulfate reduction as well."

We will additionally include clearer language in the conclusion that summarizes these points.

Referee 2: The manuscript is very lengthy and needs to be streamlined. In particular, the methods and results can be shortened drastically. Some of the language is not suitable for scientific writing, i.e., no. 232"killed", no. 362 "ghostly", no. 548 "front-line in a battle", and no. 763 "locked up".

Authors' Response:  Thank you for your input. We will trim down the methods by removing unnecessary detail  (see more detailed list of planned changes below). Note though that by the creation of a 'Study Site' chapter, as recommended by both Referees, the method section will slightly gain length elsewhere. We will further remove the language that is considered not suitable for scientific writing.

Referee 2: Please double check all unit conversions as well as there appear to be some discrepancies.

Authors' Response:  Thank you for bringing this to our attention. As also detected by Referee 1, there was information lacking for the conversion of units in the calculation of sulfate reduction rates. However, this section will be removed to shorten the method section since we followed exactly the protocol of Kallmeyer et al. 2004. We will also double-check all other conversions.

Referee 2:
Introduction
The narrative of the introduction is quite difficult to follow and needs to be restructured. I suggest that the authors move the section describing the Santa Barbara Basin to the methods.

Authors' Response:  Thank you for your suggestion, we will move the second paragraph of the introduction into a separate "study site" section within the Methods.

Referee 2:
Methods & Results
The methods and results are quite lengthy and need to be streamlined. Some redundant statements include but are not limited to: no. 158–159: "Station depth, latitude, and longitude were automatically…", no. 167: "Cores when then stored in a 6-core capacity basket…", and no. 190–191:"Geochemical analyses were… …organic matter degradation".

Authors' Response:  Thank you for your input. We will trim down the methods by removing unnecessary details (e.g., details of the underwater elevator used). We will trim down the results section of the revised version by reducing redundancy (e.g., duplicative sediment lamination descriptions), removing unnecessary detail (e.g., referring to fewer stations when giving examples), and removing unnecessary descriptions of data (e.g., changes between core supernatant and porewater solute concentrations).

Referee 2: no. 179: Were the pore water samples filtered during or after centrifugation? If not, then it is possible that particles and colloids are present (e.g., Raiswell and Canfield, 2012 Geochemical Perspectives). This should be explained in more detail as it influences the measured concentrations of the different species.

Authors' Response: Thank you for this suggestion. We did not filter the porewater after sediment centrifugation, as we did not perform this work (including sediment core slicing) inside an anoxic glove bag. Instead, we subsampled sediment directly into argon-filled centrifuge vials under an argon flow. The advantage of this procedure is a much faster handling of sensitive species, which we prefer over lengthy core slicing in a glove bag. We will mention in section 2.3 (Sediment Core Sub-Sampling) and section 2.4 (Sediment Porewater Geochemistry) that sediment porewater was unfiltered. Since filterable iron appears to be the most important constituent affected, we will also include a few sentences in Discussion section 4.3 highlighting that the unfiltered porewater contains colloidal and nanoparticulate $Fe^{2+}$, but that it was unlikely to affect the order of magnitude of the measured porewater. A tentative draft of this text reads:

"It should be noted that porewater samples for geochemical analyses were unfiltered and hence reported iron concentrations include aqueous, colloidal, and nanoparticulate species. Irrespective, all these components represent bioavailable sources of iron. Further, since filtering through 0.45 or 0.2 µm filters only removes a fraction of colloidal particles and no nanoparticles (Raiswell and Canfield, 2012), potential surplus porewater iron in SBB samples compared to other studies was likely minimal."

Referee 2: no. 199: "DIC detection limit was 0.5 mM" Why is the detection limit so high? Is this possibly a typo?

Author's Response:  We reported a detection limit of 0.5 mmol/L as this was the concentration of the lowest standard that was used to build the calibration curve. However, we will report the methodological limit of 0.1 mmol/L as reported in Hall and Aller (1992) to avoid confusion. Thanks!

Discussion
The findings need to be discussed in more context. Overall, I believe that the discussion will benefit from a schematic overview that illustrates the environmental conditions controlling the proliferation of giant sulphur-oxidizing bacterial mats. Several statements are being made without explaining the underlying mechanisms (see specific comments below).

Author's Response:  Thank you for your feedback. We agree that this study would benefit from a schematic overview and plan to add one to the revised manuscript. We will re-organize and reword the conclusion section to provide an overview of the changes to GSOB mats and underlying sediments upon basin deoxygenation. We will also add language throughout the discussion to clarify the underlying mechanisms involved in mat proliferation. Specifically, we will add a thermodynamic discussion of why DNRA is favored at lower nitrate concentrations and how GSOB could create those conditions by hyper-accumulating nitrate into their vacuoles. We will also clarify that this hyper-accumulation could starve the sediment-water interface of terminal electron acceptors, allowing for the exhaustion of iron oxides in the upper sediment, and encourage the rise of the sulfate reduction zone. We hope this provides sufficient additional context for our findings. More detailed comments and tentative drafts of the revised text are provided below.

no. 595–613: What are the underlying mechanisms behind low nitrate availability controlling the proliferation of giant-sulphur oxidizing bacteria? Following the pre-established notion that giant-sulphur oxidizing bacteria can use nitrate as an electron acceptor, this would suggest that a higher nitrate availability, in fact, enhances proliferation of giant-sulphur oxidizing bacteria. Is there a critical threshold, that optimizes growth? This needs to be discussed in further detail.

Authors' Response: Thank you for the suggestion. We will include more precise citations and in-text references to studies that have examined the thermodynamics between DNRA and denitrification at different electron donor to acceptor ratios. We will further add a table (Table 3) with the Gibbs free energies for metabolisms relevant (DNRA with nitrate/sulfide and denitrification with nitrate/TOC) for this discussion. We are further working on integrating a calculation of the Gibbs free energy of denitrification coupled to acetate oxidation based on acetate concentrations determined in a study recently accepted in Biogeosciences, which was done in tandem with our work (Krause et al., in press).

A tentative draft of the text reads:

 "A high ratio of electron donor to electron acceptor favors DNRA over denitrification  (Marchant et al., 2014; Hardison et al., 2015; Tiedje et al., 1983) and this ratio appears to be critical in determining the dominant nitrate reduction pathway in SBB sediments, similar to the Eastern Gotland Basin (Hylén et al., 2022) and the By Fjord (De Brabandere et al., 2015). The energy yields for denitrification and DNRA at environmental conditions in the SBB depocenter (6°C, 58 atm) are shown in Table 3.  As discussed in (Tiedje et al., 1983), heterotrophic denitrification yields more energy per mol of electron donor than DNRA. However, the reverse is true when considering energy yield per mol of electron acceptor ($NO_3^-$). DNRA also yields 3 more electrons per molecule of $NO_3^-$ than denitrification. Tiedje et al. argued that in environments that are starved of powerful terminal electron acceptors, (e.g., oxygen, nitrate) such as anoxic, organic-rich sediment, the energy yield per electron acceptor and additional electrons available

for transfer, could push nitrate reduction towards DNRA. Multiple laboratory studies have converged on an electron donor to acceptor ratio of approx. 3 to encourage DNRA over denitrification (Hardison et al., 2015; Algar and Vallino, 2014) though other studies have found higher values (Porubsky et al., 2009; Kraft et al., 2014). Our TOC results (Table 2) suggest that bottom water nitrate concentrations would need to drop to approx. 2 µM before DNRA would be more favorable, however, the sulfide concentrations near the sediment-water interface at the SBB depocenter (approx. 200 µM at 0.5 cm depth; Fig. 3, NDRO) would favor chemoautotrophic DNRA over denitrification at ambient marine nitrate concentrations (approx. 28 µM). Additionally, DNRA appears to be the preferred nitrate reduction pathway for chemoautotrophs that utilize iron or sulfide as an electron donor (Caffrey et al., 2019; Kessler et al., 2019; An and Gardner, 2002). As GSOB mats hyper-accumulate nitrate from the bottom water into their intracellular vacuoles, the resulting decline in electron acceptors at the sediment-water interface coupled with an elevation of the sulfate reduction zone would create an electron donor to acceptor ratio that favors DNRA. Since GSOB mats in the SBB seem to prefer DNRA, starving the bottom water of electron acceptors coupled with the high sulfate reduction rates afforded by the sediment TOC content could give them a competitive advantage and allow them to proliferate into the largest-yet mapped GSOB mat in Earth's oceans, as seen in other expeditions (Valentine et al., 2016; Reimers et al., 1996b; Kuwabara et al., 1999)."

Tiedje, J. M., Sexstone, A. J., Myrold, D. D., and Robinson, J. A.: Denitrification: ecological niches, competition and survival, Antonie van Leeuwenhoek, 48, 569-583, 1983.

Krause, S. J. E., Liu, J., Yousavich, D. J., Robinson, D., Hoyt, D., Qin, Q., Wenzhoefer, F., Janssen, F., Valentine, D., and Treude, T.: Evidence of cryptic methane cycling and non-methanogenic methylamine consumption in the sulfate-reducing zone of sediment in the Santa Barbara Basin, California, EGUsphere [preprint], https://doi.org/10.5194/egusphere-2023-909, 2023.

Referee 2: no. 636–637: Benthic release of toxic sulphide into the bottom water is the real culprit. So perhaps it is better to highlight this instead of mentioning accumulation of sulphide at the sediment-water interface.

Authors' Response:  Thank you for the suggestion. This section of our discussion is talking about toxicity specifically towards GSOB, so the sediment-water interface is pertinent here. However, the reviewer points out that we do not discuss how sulfur-oxidizing bacteria prevent water column euxinia in the SBB. We will add a sentence within this section 4.4 to address that.

Referee 2: no. 665–678: What is the relationship behind the formation of iron sulphide in deeper layers and the depth of the sulphate reduction zone? This needs to be discussed in more detail.

Authors' Response: We will rephrase this sentence to emphasize that the exhaustion of iron oxides in upper sediments, rather than iron sulfide formation, is required to elevate the zone of sulfate reduction in the SBB. We will then detail the patterns of porewater $Fe^{2+}$ and sulfate reduction throughout the basin to provide evidence for this hypothesis. A tentative draft of the text reads:

"The hyper-accumulation of NO3- by GSOB mats potentially facilitates sulfate reduction close to the sediment-water interface in the SBB (e.g., NDRO and NDT3-A as seen in fig. 2N and 2O) by starving the sediment of this more powerful electron acceptor. The upward rise of the sulfate reduction zone at NDT3-C (fig. 2L) further suggests that the exhaustion of iron oxides and the formation of iron sulfide

below the sediment-water interface may play a crucial role in controlling the distribution of sulfate reduction as well."

The last paragraph of the section provides a discussion of how changes in cable bacteria density at the onset of anoxia in the SBB could allow for the exhaustion of iron oxides to take place.

Referee 2: no. 725: This is shown in an overview study by Hermans et al., 2019 (Environmental Science and Technology) where 12 sites with contrasting bottom water redox conditions are being compared. Both the activity and relative abundance is low at the anoxic sites.

Authors' Response: Thank you for pointing us to this study. We will include a citation of this paper after the statement "…they appear to be inactive in anoxic aquatic environments".

Referee 2: no. 727–729: Please be aware that the filaments are typically not stretched out vertically in the sediment. In other words, a filament with a total length of 7 cm might only span a distance of e.g., 3 cm in the sediment. This is important to keep in mind. What is the actual distance between the sediment-water interface and the appearance of iron mono-sulphides in your core?

Authors' Response: Thank you for this clarification. Iron sulfides were detected, based on visual appearance, beginning at approx. 5 cm depth. More in-depth mineralogical analyses of these sediments will be detailed in a future study. We will change the text to clarify that cable bacteria do not typically stretch this large a distance. A tentative draft of the text reads:

"The maximum recorded filament length of cable bacteria is 7 cm (Van De Velde et al., 2016), though typically they are not stretched completely vertically through the sediment. The appearance of black sediment in the SBB C-station sediments, starting at approx. 5 cm depth, could be an indication that cable bacteria are oxidizing iron sulfides at that sediment depth and prevent their formation at shallower depths."

Referee 2: no. 732: Rephrase to "prevent benthic release of sulphide" instead of "euxinic conditions at the sediment-water interface".

Authors' Response: Thanks for the suggestion! We will change the wording as requested.

Referee 2: no. 737–739: The pH peak directly below the sediment-water interface is typically absent in sediment populated by cable bacteria in field settings (as illustrated in the Supplementary Information of Hermans et al., 2019 Environmental Science and Technology). However, the distinct peak is more common under ideal incubation settings. This should be addressed.

Authors' Response: Thanks for the suggestion, we will include a caveat that cable bacteria associated pH increases are more often seen in laboratory settings but not in the field. A tentative draft of the text reads:

"Cable bacteria can sometimes be detected in sediments via a slight pH increase (typically pH > 8) (Schauer et al., 2014) which was not reflected in our pH results, but this phenomenon is more typically seen in the laboratory and not the field (Hermans et al., 2019)."

Referee 2: no. 748–749: This has been shown in previous studies (e.g., Mortimer, 1941 Journal of Ecology). Furthermore, the preferential regeneration of phosphorus from organic matter under deoxygenated bottom waters further enhance benthic phosphorus release (e.g. Van Cappellen & Ingall, 1994 Paleoceanography).

Reply:  Thank you for the suggestion. We will include a sentence at the beginning of this paragraph highlighting both studies. Additionally, we will clarify that high phosphate flux from anoxic sediments is due to both the reduction of iron oxides, as well as the release of phosphate from organic matter. A tentative draft of the text reads:

"Release of dissolved iron and phosphate from sediment below anoxic waters is a well-documented phenomenon (e.g., (Mortimer, 1941; Van Cappellen and Ingall, 1994; Van De Velde et al., 2020; Noffke et al., 2012)) and this phenomenon is seen in the SBB as well. As postulated previously (Kuwabara et al., 1999), basin flushing oxidizes iron sulfides at the sediment-water interface, providing ample substrate for microbial iron reduction once anoxia returns. This iron reduction initiates high rates of $Fe^{2+}$ release from SBB depocenter sediment (Fig. 5). Iron reduction further releases iron-bound $PO_4^{3-}$ (Mortimer, 1941) as seen by high benthic fluxes of $PO_4^{3-}$ at the depocenter (Fig. 5), although notably some of this $PO_4^{3-}$ release could be also attributed to organic matter degradation (Van Cappellen and Ingall, 1994)."

---

## Author Response (AR2)

Dear Caroline Slomp,

Thank you very much for your help and additional input. We have expanded the discussion on eukaryotic denitrification to include more detail about radiolarians in the Santa Barbara Basin, and quoted the rates of denitrification found in Bernhard et al. 2012. We added an additional sentence discussing denitrification by fungi which we believe could also be a possibility in the Santa Barbara Basin. This was a great addition to our discussion, and we are excited to have it included.

Beyond your suggested edits, we discovered an error with the units for DIC and alkalinity in Figures 2 & 3 and have corrected that (replaced µM by mM). Additionally, we re-exported figures 1, 4 & 5 to increase the image quality and we changed the font for Figures 1A and 1B to be consistent with the rest of the manuscript. Lastly, we added one person to the acknowledgements.

Thank you!

David Yousavich and Tina Treude in the name of all Co-Authors